**RESEARCH**

# Genetic regulation of RNA splicing in human pancreatic islets

Goutham Atla[1,2,3], Silvia Bonàs-Guarch[1,2,3]*, Mirabai Cuenca-Ardura[1,2], Anthony Beucher[1,2,3], Daniel J. M. Crouch[4], Javier Garcia-Hurtado[1,2], Ignasi Moran[3,5], the T2DSystems Consortium, Manuel Irimia[1], Rashmi B. Prasad[6,7], Anna L. Gloyn[8,9], Lorella Marselli[10], Mara Suleiman[10], Thierry Berney[11], Eelco J. P. de Koning[12,13], Julie Kerr-Conte[14], Francois Pattou[14], John A. Todd[4], Lorenzo Piemonti[15] and Jorge Ferrer[1,2,3]*

*Correspondence:
silvia.bonas@crg.eu; jorge.
ferrer@crg.eu

[1] Centre for Genomic Regulation,
The Barcelona Institute
of Science and Technology,
Barcelona, Spain
Full list of author information is
available at the end of the article

## Abstract

**Background:** Non-coding genetic variants that influence gene transcription in pancreatic islets play a major role in the susceptibility to type 2 diabetes (T2D), and likely also contribute to type 1 diabetes (T1D) risk. For many loci, however, the mechanisms through which non-coding variants influence diabetes susceptibility are unknown.

**Results:** We examine splicing QTLs (sQTLs) in pancreatic islets from 399 human donors and observe that common genetic variation has a widespread influence on the splicing of genes with established roles in islet biology and diabetes. In parallel, we profile expression QTLs (eQTLs) and use transcriptome-wide association as well as genetic co-localization studies to assign islet sQTLs or eQTLs to T2D and T1D susceptibility signals, many of which lack candidate effector genes. This analysis reveals biologically plausible mechanisms, including the association of T2D with an sQTL that creates a nonsense isoform in *ERO1B*, a regulator of ER-stress and proinsulin biosynthesis. The expanded list of T2D risk effector genes reveals overrepresented pathways, including regulators of G-protein-mediated cAMP production. The analysis of sQTLs also reveals candidate effector genes for T1D susceptibility such as *DCLRE1B*, a senescence regulator, and lncRNA *MEG3*.

**Conclusions:** These data expose widespread effects of common genetic variants on RNA splicing in pancreatic islets. The results support a role for splicing variation in diabetes susceptibility, and offer a new set of genetic targets with potential therapeutic benefit.

**Keywords:** RNA splicing, Type 1 diabetes, Type 2 diabetes, TWAS, G-protein signaling, Pancreatic islets, Beta cells, Senescence, CTRB2, Pancreatic beta-cells, Diabetes pathophysiology, Quantitative trait loci

## Background

Genome-wide association studies have identified hundreds of genomic loci that carry genetic variants contributing to type 2 diabetes (T2D) and type 1 diabetes (T1D) susceptibility [1–5]. The vast majority of associated genetic variants are non-coding, and epigenomic studies have revealed that many are located in human pancreatic islet transcriptional cis-regulatory elements [6–9]. Numerous T2D risk loci have thus been assigned to effector transcripts through human islet expression quantitative trait loci (eQTLs), three-dimensional chromatin maps, and genome editing experiments [10–13]. Such studies have established that gene expression variation in pancreatic islets is critically important for T2D susceptibility. A subset of T1D susceptibility signals have also been proposed to act through expression variation in islet cells, sometimes involving shared genes with T2D [14]. A large fraction of T2D and T1D risk loci, however, cannot be ascribed to transcriptional regulatory mechanisms in pancreatic islets or other tissues, pointing to additional non-coding mechanisms that remain to be defined.

Alternative splicing of pre-mRNAs provides an additional mechanism whereby genetic variation can create functional diversity across human genomes. Rare mutations and common variants that influence pre-mRNA splicing have been linked to a broad range of human diseases [15–17]. The effects of genetic variants on pre-mRNA splicing in human pancreatic islets, however, are largely unexplored. GTEx, the most comprehensive catalogue of splicing QTLs (sQTLs) across human tissues, does not include human pancreatic islets [17]. Other studies have examined exon-level expression in islet RNAs [12, 18, 19], although this does not capture the complexity of alternative splicing and is confounded by total gene expression variation. We have now directly examined RNA splicing in a panel of 399 human islet samples and created an atlas of sQTLs. This showed that sQTLs impact key genes for islet biology and diabetes. We uncover sQTLs that show fine colocalization with genetic variants associated with T2D, discover new genetic associations, and expand the spectrum of putative gene effectors of disease susceptibility. We further reveal new candidate mediators of T1D susceptibility. These findings provide biological insights into genetic mechanisms underlying diabetes risk.

## Results

To examine the impact of common genetic variation on RNA splicing in pancreatic islets, we aggregated RNA-seq and genotype data from 447 human pancreatic islet samples from four cohorts [11, 13, 18, 19] and processed them to yield 399 qualifying samples after applying genotype and RNA-seq quality controls (Fig. 1a). We corrected for known and unknown covariates (Additional file 1: Fig. S1) and performed QTL analysis on mRNA levels (eQTLs) and junction usage [20] (sQTLs), using 6.46 million high-quality common variants (Additional file 1: Fig. S2a). Focusing on genes expressed in >10% of samples in each cohort, we found *cis*-eQTLs in 3433 genes (eGenes) at FDR ≤ 1% (Fig. 1a, Additional file 2: Table S1 [21]). This analysis also revealed a widespread impact of genetic variants on splicing variation, with 4858 *cis*-sQTLs at FDR ≤ 1%, 25% of which showed >10% shift in splice site usage in reference vs. alternate alleles (Fig. 1a, b, Additional file 3: Table S2 [21]). The 4858 sQTL junctions included alternative usage of 5′ exons, 3′ exons, mutually exclusive or skipped exons, or influenced combinations of

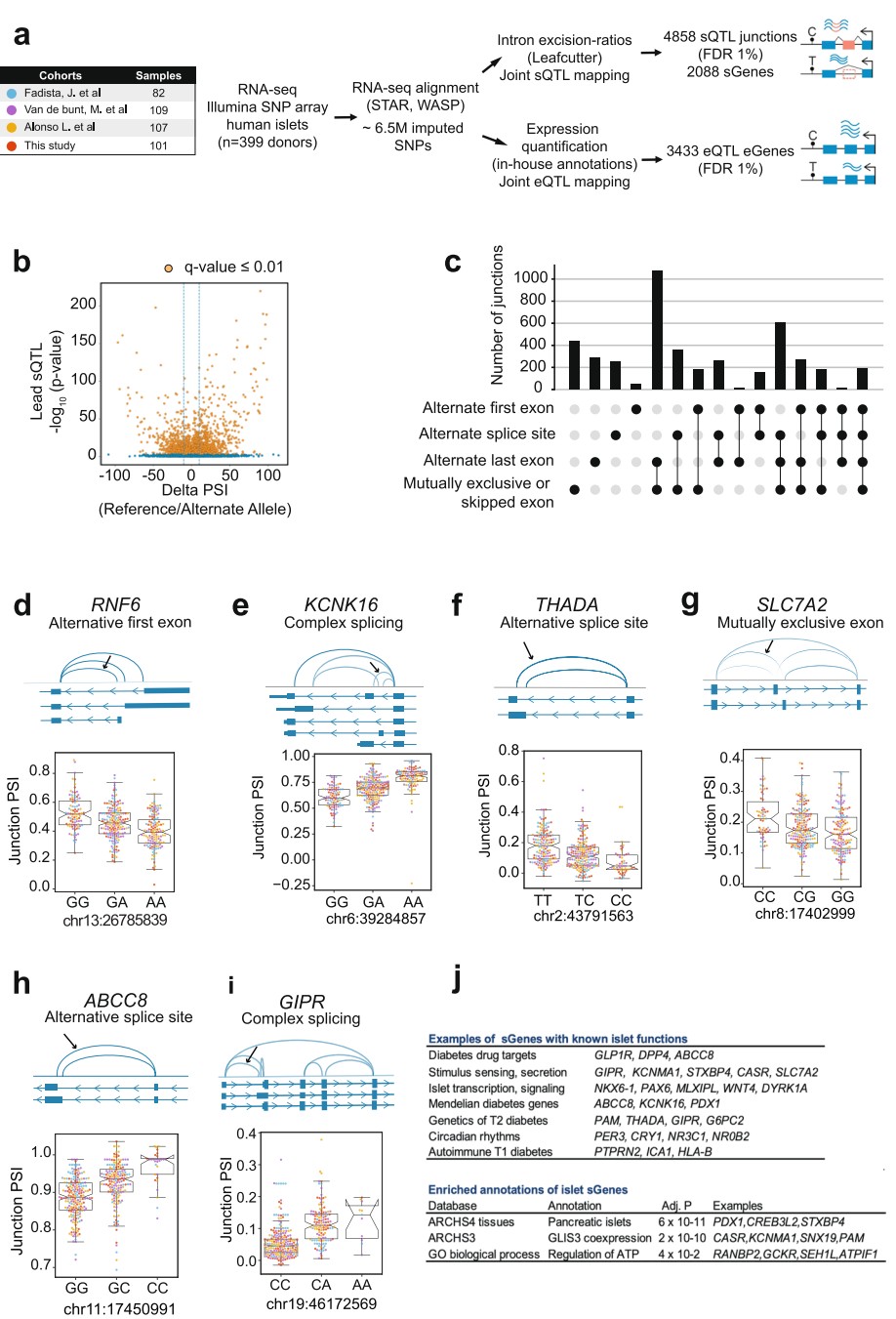

**Fig. 1** Mapping sQTLs and eQTLs in human pancreatic islets. **a** Overview of the study design. **b** Volcano plot showing the reference to alternate allele change in percentage splice index (Delta-PSI) for junctions, and sQTLs -$\log_{10}$ *p*-values. Orange dots depict sQTLs junctions with $q \leq 0.01$. **c** Classification of sQTLs according to types of splicing events. **d–i** Selected examples of sGenes with different types of splicing events. An arrow signals the sQTL junction with best *p*-value, and adjacent boxplots show normalized, batch-corrected junction PSI values stratified by the lead sQTL genotype (IQR and 1.5 × IQR whiskers). Junction PSI values are colored according to the human islet dataset they belong to (see **a**). All boxplots show sQTLs with permutation *p*-values significant at FDR ≤ 1%, see Additional file 3: Table S2. **j** Functional annotations of sGenes. The top panel shows a manually curated list of examples with known functions in islet function and diabetes (see Additional file 4: Table S3); the bottom panel shows enriched annotations using EnrichR and Benjamini–Hochberg-adjusted *p*-values

such splice variants (Fig. 1c–i). The junctions mapped to 2088 distinct genes (sGenes), of which ~90% were known protein-coding genes, and ~7% were lncRNAs (Additional file 1: Fig. S2b).

We benchmarked human islet splice variants against GENCODE [22] and other available transcriptome maps and found that only 77% of the sQTL junctions were annotated. This overlap increased to 90% in comparisons with unpublished human islet transcript annotations built with long-reads (Atla, G., Beucher, A., et al., unpublished) (Additional file 1: Fig. S2c). This suggests that human islet transcripts are still incompletely annotated, but nevertheless well captured by our analysis of splice junctions.

We compared islet sQTLs to previously reported exon-QTLs [12], a possible proxy for sQTLs, and found that only 18% of sQTL junctions were flanked by exons from exon-QTLs. Furthermore, when sQTL and exon-QTLs affected the same gene, there was limited linkage disequilibrium ($r^2 < 0.6$) between the lead sQTL and exon-QTL variant for 45.2% of overlapping genes (Additional file 1: Fig. S2d, e). This indicates that sQTLs, which directly measure splice junction variation, and exon-QTLs, which measure exon expression levels and can thus be influenced by variables unrelated to RNA splicing, mostly capture fundamentally different events.

Islet sGenes were enriched in islet-specific co-expression networks (Fig. 1j) and included numerous genes with well-established roles in islet biology and diabetes (Additional file 4: Table S3), including major drug targets (*GLP1R*, *DPP4*, *ABCC8*), regulators of hormone secretion (*SLC7A2, CASR, GIPR*), transcription (*NKX6-1, PAX6, MLXIPL/ChREBP*), signaling (*DYRK1A, WNT4*), or circadian rhythms (*PER3, CRY1, NR3C1, NR0B2*). Importantly, sQTLs also affected genes that harbor variants that cause monogenic diabetes (*KCNK16, ABCC8, PDX1*) or influence T2D risk (*THADA, PAM*) as well as genes that play a role in T1D pathogenesis (*ICA1, PTPRN2, HLA-B*) (Fig. 1d–j, Additional file 1: Fig. S3, Additional files 3 and 4: Table S2-S3). Our results, therefore, disclosed a pervasive impact of common genetic variants on alternative splicing of human pancreatic islet transcripts, including numerous genes that are important for islet function and differentiation, diabetes treatment, or pathophysiology.

### sQTLs and eQTLs are distinct

We next explored the extent to which genetic effects on splicing and expression of islet transcripts were distinct. Only 34% of genes that harbored sQTLs (715 sGenes) also harbored a significant eQTL (Fig. 2a). Furthermore, for those 715 common genes, the lead eQTL and sQTL were frequently not in linkage disequilibrium ($r^2$ <0.6 for 57% of genes, <0.1 for 24% of genes, Fig. 2b). Thus, in most genes that harbored both eQTLs and sQTLs, these were driven by independent signals. This is illustrated by *RGS1*, encoding a regulator of G-protein signaling, which has independent variants affecting either mRNA expression or exon skipping (Fig. 2c). In keeping with these findings, eQTLs and sQTLs were enriched in different functional genomic annotations. sQTLs were predominantly enriched in 5′ or 3′ splice sites and exons, whereas eQTLs showed a predominant enrichment in active promoters and enhancers (Fig. 2d, Additional file 1: Fig. S2f, Additional files 5 and 6: Table S4-S5). We also found differences in the extent to which genetic effects on splicing or expression differed across tissues; ~60% of lead islet sQTLs showed significant sQTLs in <5 GTEx tissues [17], compared to ~30% of lead eQTLs, suggesting

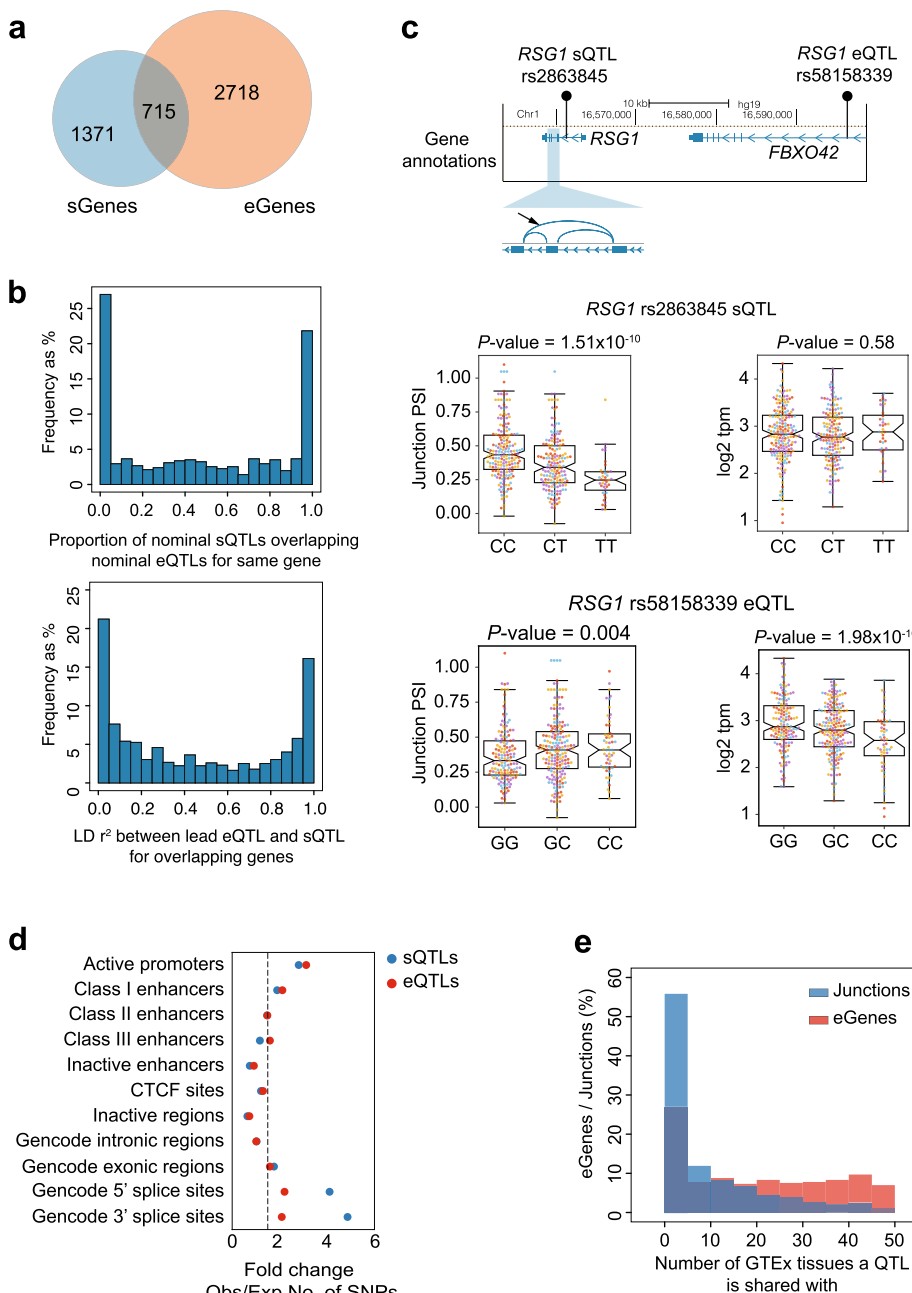

**Fig. 2.** sQTLs and eQTLs are distinct genetic signals. **a** Overlap of sGenes and eGenes. **b** For 715 genes that have both eQTLs and sQTLs (overlapping genes in **a**), the top histogram shows the distribution of the percentage of variants shared between sets of nominally significant eQTLs and sQTLs. The bottom histogram shows the distribution of LD ($r^2$) values between the lead eQTL and sQTL. **c** *RSG1* has a distal eQTL, located in an intron of the *FBXO42* gene, and an intronic sQTL, both of which are in low LD ($r^2$=0.25). Boxplots represent *RSG1* expression and junction PSI values for both sQTL and eQTL, showing that the lead eQTL rs58158339 is not associated with *RSG1* splicing and the sQTL rs2863845 is not associated with expression. Boxplots show normalized, batch-corrected expression or junction PSI values stratified by the genotype of the lead QTL (IQR boxes and 1.5 × IQR whiskers). Individual samples are colored according to the human islet dataset they belong to (see color legend in Fig. 1a). Nominal QTL *p*-values are provided. **d** Enrichment of sQTL and eQTL variants in different functional genomic annotations using GREGOR. The *x*-axis represents GREGOR fold change of observed vs. expected number of SNPs at each functional annotation. The dotted line represents 1.5-fold change. **e** Percentage of eGenes and Junctions with eQTLs or sQTLs at FDR ≤ 1%, respectively, shared in different number of GTEx tissues. We used significant eQTLs and sQTLs from 47 distinct GTEx V8 release tissues

a significant islet-specific component of sQTLs (Fig. 2e). Taken together, our results reveal two separable layers of genetic regulation of the human islet transcriptome.

### Islet sQTLs provide new T2D and glycemic trait targets

Genetic susceptibility to T2D has been linked to sequence variants that influence gene transcription in human islets [10–13], but the relationship with islet splicing has not been systematically explored. To examine the potential contribution of islet sQTLs to T2D genetic associations [1], we first used quantile-quantile plots that compare the distribution of T2D association *p*-values of sQTL and eQTL variants against an expected null distribution (Fig. 3a). As anticipated, eQTLs showed strong inflation of more significant T2D association *p*-values. Remarkably, sQTLs also showed genomic inflation of T2D risk *p*-values (Fig. 3a). Importantly, this effect was maintained with islet-selective sQTLs (junctions with sQTLs in ≤ 5 GTEx tissues), or after omission of sQTLs that had high linkage disequilibrium with eQTLs (lead sQTL is in $r^2 \geq 0.6$ with the lead eQTL for the same gene) (Additional file 1: Fig. S4a, b). Furthermore, sQTLs also showed genomic inflation of low association *p*-values with T2D-related traits such as fasting glycemia (FG) or fasting insulin (FI) [23] (Additional file 1: Fig. S4c, d). These observations, therefore, suggest that splicing variation in human islets could contribute to T2D genetic susceptibility.

We reasoned that if splicing variation is linked to disease susceptibility at specific loci, a fraction of sQTLs and T2D associations should show high colocalization evidence (posterior probability of shared association between both phenotypes ≥ 0.8), and this could in turn point to specific candidate effector or causal transcripts underlying disease pathophysiology. To this end, we performed a systematic colocalization analysis between our islet sQTL or eQTLs, and independent GWAS signals for T2D ($n = 403$) [1] or glycemic traits (fasting glycemia, FG/fasting insulin, FI) ($n = 274$) [23]. We applied colocalization as implemented in *gwas-pw* [24], which draws upon the original *coloc* algorithm but does not rely on user-defined priors. This identified candidate effector transcripts with high colocalization support (posterior probability of shared association between both phenotypes ≥ 0.8) at 9 independent T2D GWAS signals using sQTLs, and 25 using eQTLs (Additional files 7 and 8: Table S6-S7, Additional file 1: Fig. S6a). At loci

(See figure on next page.)

**Fig. 3** Role of human islet splicing variation in T2D susceptibility. **a** Quantile-quantile (QQ) plot showing observed T2D association *p*-values in human islet sQTLs (orange dots) and eQTLs (blue dots) against *p*-values under the null hypothesis. The grey-shaded region represents 1000 *p*-value distributions (in the -log$_{10}$ scale) of random sets of control sQTL variants (see the "Methods" section). Each set of control variants matches the number of islet sQTLs plotted. **b** Manhattan plot of splicing associations with T2D susceptibility (sTWAS). The *y*-axis shows -log$_{10}$ TWAS association *p*-values. Significant sTWAS associations in known T2D GWAS loci are colored in purple, and in previously unreported loci in orange. **c**, **d** Regional T2D GWAS signal plots for *PTPN9* and *PKLR* loci, two known T2D susceptibility regions, which showed significant sTWAS signals in *MAN2C1* (**c**) and *SCAMP3* (**d**) genes, respectively. Both splicing QTL effects in *MAN2C1* and *SCAMP3* are significant at FDR ≤ 1%; see Additional file 3: Table S2. LocusZoom plots show -log$_{10}$ T2D association *p*-values and locations in hg19 genome build. LocusCompare scatter plots show sQTL and T2D GWAS association *p*-values (-log$_{10}$ scale), illustrating co-localization of variants for both traits. Variants are colored according to the LD correlation ($r^2$) with the lead GWAS variant of the sTWAS association (purple diamond). Boxplots show normalized, batch-corrected junction PSI values on *y*-axis stratified by the genotype of the lead GWAS variant from the sTWAS association. Boxplots follow the color-to-batch legend from Fig. 1a. **e** Known T2D loci with target effector transcripts nominated by sQTLs and eQTLs from this study and/or eQTL maps from the InsPIRE consortium

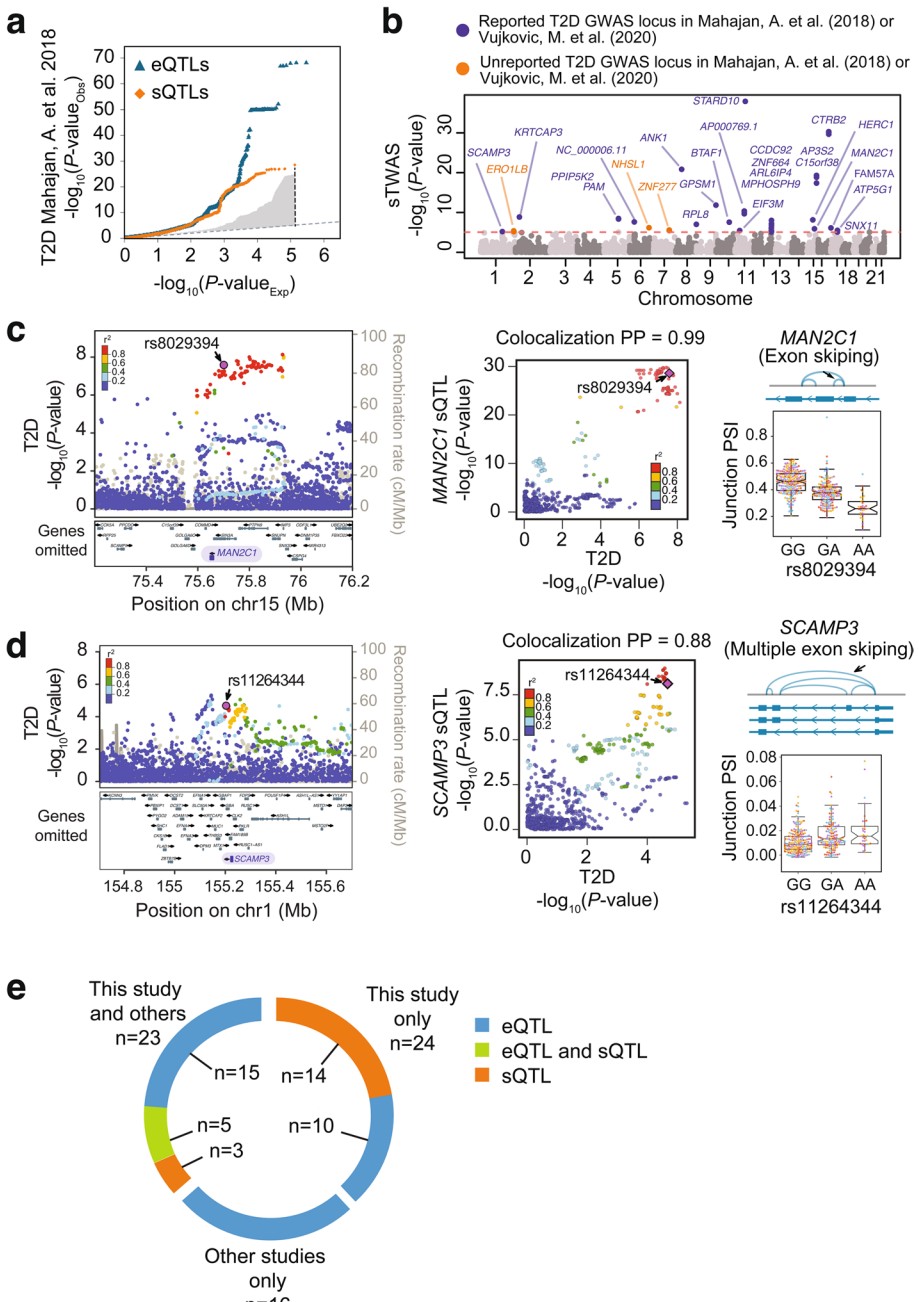

**Fig. 3** (See legend on previous page.)

associated with glycemic traits, we found colocalization with sQTLs for eight putative target genes, and eQTLs for 17 genes (Additional files 7 and 8: Table S6-S7).

We further harnessed Transcriptome-Wide Association Studies (TWAS) to expand the collection of candidate effector genes for T2D and related traits. In this approach, which so far has not been applied to human islet RNA-seq data, genetic effects on splicing or expression are used to impute transcript variation in cases versus controls from GWAS datasets. This allowed us to identify splicing or expression changes associated with T2D and related traits. More specifically, we used the FUSION algorithm [25] and

GWAS summary statistics [1] to identify T2D associations with islet junction usage or gene expression (sTWAS and eTWAS, respectively). Because TWAS findings do not necessarily distinguish between shared genetic effects and linkage [26, 27], we focused on TWAS signals showing colocalization with the GWAS phenotype (PP4 $\geq$ 0.6), thus minimizing confounding effects from linkage. This identified 27 genes (42 splicing events) showing significant sTWAS with T2D risk, and 29 genes with eTWAS after multiple testing correction (Bonferroni $p= 8.6 \times 10^{-6}$ and $1.8 \times 10^{-5}$ after correcting for 5804 splicing junctions and 2851 genes, respectively) (Fig. 3b, Additional file 1: Fig. S4e, S5, Additional files 9 and 10: Table S8-S9). For glycemic traits, we identified 22 candidate target genes (43 splicing junctions) of GWAS signals via sTWAS, and 16 candidate target genes via eTWAS (Additional file 1: Fig. S4f-i, Additional files 9 and 10: Table S8-S9).

As expected, most TWAS signals fell in loci showing significant T2D and glycemic trait associations in GWAS (Fig. 3b, Additional file 1: Fig. S4e-i, S6b, Additional files 9 and 10: Table S8-S9). This included a sTWAS association for *MAN2C1*, encoding α-Mannosidase that has been implicated in mitochondrial-induced apoptosis and tissue damage [28, 29] (Fig. 3c). However, sTWAS revealed six T2D associations that did not reach genome-wide significance in the reference GWAS [1], three of which (*SCAMP3*, *SNX11*, and *FAM57A*) were nevertheless significant in a recent trans-ancestral meta-analysis for T2D [3] (Additional file 9: Table S8). *SCAMP3* encodes a vesicular transport protein [30] with unknown function in islets (Fig. 3d). Other significant sTWAS for T2D did not reach significance in GWAS reported so far, namely those encoding *ERO1B*, *NHSL1*, and *ZNF277* (Fig. 3b, Additional file 9: Table S8; further details for *ERO1B* are shown in Fig. 4g). Thus, sTWAS nominated putative effector targets for T2D and glycemic trait genetic associations, and identified additional genetic associations.

Our studies also highlighted significant eTWAS for T2D, including *PCBD1*, which is mutated in a syndrome that includes monogenic diabetes [31, 32], and encodes a co-factor of HNF1A, an established monogenic diabetes gene [33] (Additional file 1: Fig. S6c, Additional file 10: Table S9). This locus was not significant in the reference GWAS [1] but another recent meta-analysis detected a significant association in this locus [3]. Our findings indicate that *PCBD1* is a strong candidate effector transcript for this T2D susceptibility locus. Previously unreported T2D genetic association signals were found through eTWAS for *CTC-228N24.2* and *VSNL1*, a calcium sensor that modulates cAMP and insulin secretion [34] (Additional file 1: Fig. S6d, Additional file 10: Table S9).

Taken together, the combination of our TWAS and QTL colocalization results identified candidate effector transcripts for 47 known T2D susceptibility loci, including 22 acting through sQTLs, and 30 through eQTLs. Out of these 47 loci, 24 lacked information about likely effector transcripts in a recent large-scale islet eQTL analysis [12]. In aggregate, our sQTL and eQTL analysis increases the number of T2D loci with candidate effector genes supported by molecular QTLs to 63, a 1.6-fold change relative to the largest islet eQTL analysis so far [12] (Fig. 3e, Additional file 11: Table S10). Likewise, the integration of our molecular QTLs and GWAS summary statistics data of glycemic traits from a recent trans-ancestral meta-analysis [23] allowed us to increase the number of GWAS loci with candidate effector genes up to 43, a 3.1-fold increase relative to the largest islet eQTL mapping study so far (Additional file 12: Table S11).

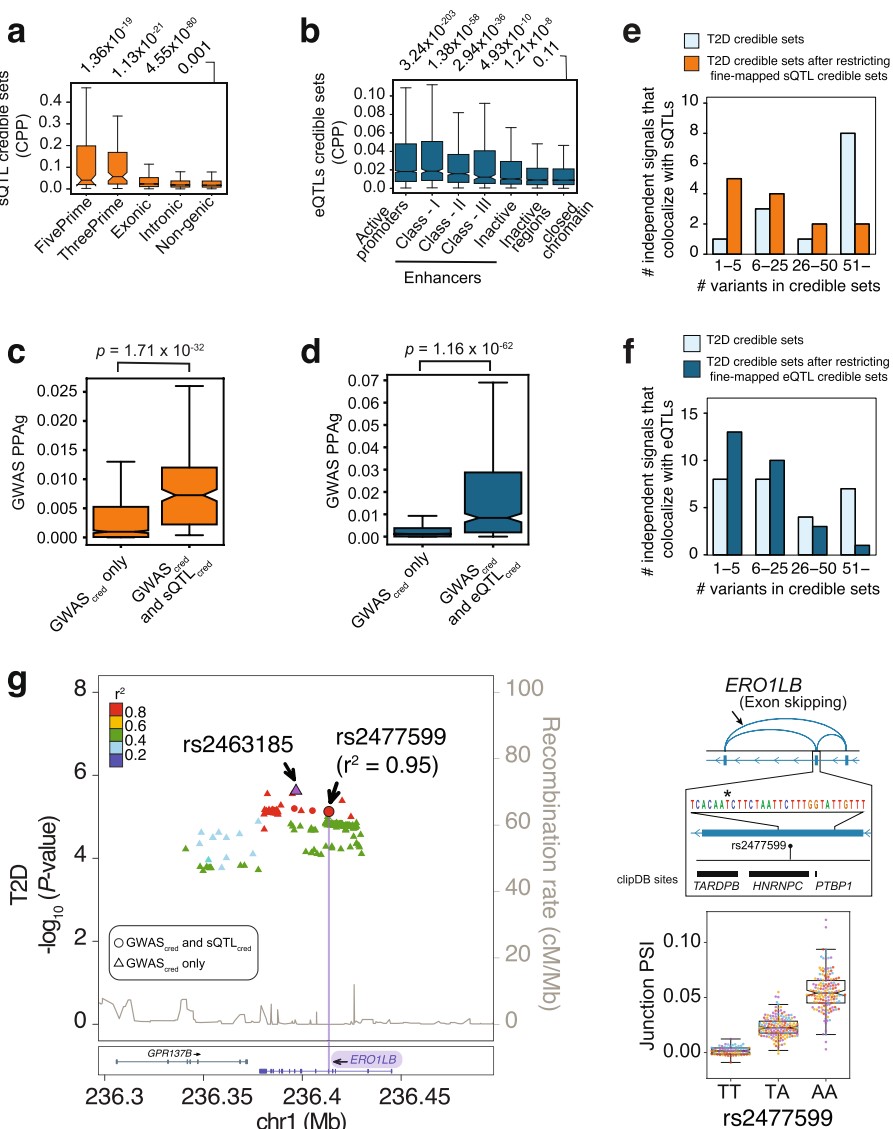

**Fig. 4** Fine-mapping causal variants for known and novel T2D genetic associations. **a** Distribution of sQTL causal posterior probabilities (CPP) across different genic and non-genic regions. *P*-values on top correspond to Mann-Whitney comparisons with non-genic regions. **b** eQTL causal posterior probabilities across epigenomic annotations. *P*-values on top correspond to comparisons with credible set variants that fall outside islet epigenomic annotations (*closed* chomatin regions). **c**, **d** For all T2D-associated loci that colocalize with an islet QTL, we examined all fine-mapped variants (99% credible sets in GWAS, $GWAS_{cred}$) and compared the distribution of T2D causal posterior probabilities for variants that are also fine-mapped QTL variants ($QTL_{cred}$) vs. those that were not fine-mapped QTL variants. Mann-Whitney *p*-values are provided. Boxplots show IQR without outliers although *p*-values were calculated using all data points. **e, f** Integration of T2D GWAS credible set variants with credible sets from colocalizing sQTLs and eQTLs increases fine mapping resolution. Bar plots show the number of independent signals that fall into different bins of number of candidate causal variants before and after restricting for QTL variants. **G** Fine-mapping an sQTL and T2D association in *ERO1B*. The splicing QTL effect on *ERO1B* is significant at FDR ≤ 1%; see Additional file 3: Table S2. The LocusZoom shows T2D association -$\log_{10}$ *p*-values; credible set variants for GWAS and sQTLs are shown as circles, and other GWAS credible set variants as triangles. The color of dots reflects $r^2$ with the lead GWAS variant (in purple) and includes the best fine-mapped candidate causal sQTL (rs2477599). The bottom inset depicts the alternative splicing event, along with the candidate causal sQTL variant and clipDB RBP binding sites. Boxplots are as described in Figs. 1, 2, and 3.

### Splicing variants uncover candidate mediators of T1D risk

We next explored the role of islet splicing variation in T1D. Despite the fact that T1D is an autoimmune disease, several observations indicate that a fraction of T1D risk alleles act through β-cells or exocrine pancreatic cells, rather than directly through immune cells [5, 14, 35]. Furthermore, significant changes in RNA splicing have been reported upon cytokine-stimulation of human islets [36]. We observed that both islet eQTLs and sQTLs show inflation of more significant T1D association values [5] (Additional file 1: Fig. S7a), and identified eQTLs and sQTLs that exhibit significant associations with T1D (Additional file 1: Fig. S7b, see complete list in Additional files 7, 8, 9 and 10: Table S6-S9). For example, islet sQTLs at *MEG3*, *ESYT1*, and *DCLRE1B* co-localize with T1D association signals (Additional file 1: Fig. S7b-d, S8a, Additional file 9: Table S8). *ESYT1* is highly expressed in β-cells and encodes extended synaptotagmin 1 (E-Syt1), which has been implicated in insulin secretion regulation by clearing the plasma membrane of diacylglycerol [37], while *MEG3* and *DCLRE1B* have biological functions consistent with a role in autoimmune diabetes. *MEG3* is an imprinted lncRNA that is strongly expressed in human β-cells and regulates cell-specific cytokine responses [38, 39]. Downregulation of *MEG3* has been observed in islets of the murine model of T1D, the nonobese diabetic mouse (NOD) [40], while targeted inhibition increases the sensitivity of β-cells to cytokine-mediated oxidative stress [41]. Studies in T1D models and human islets have indicated that β-cell senescence contributes to autoimmune β-cell destruction, while *DCLRE1B* protects telomeres from replicative damage and prevents cellular senescence [42–44]. The *DCLRE1B* sQTL and sTWAS signal is in high local LD with one of the strongest T1D association signals at *PTPN22*, yet remains independent after conditional and joint (cojo-GCTA) variant analysis (Additional file 1: Fig. S8b-c). Taken together, the results nominate islet splicing and expression variants as potential molecular mediators of T1D predisposition.

### Fine-mapping QTLs identifies candidate causal variants for T2D risk

Genetic fine-mapping of molecular QTLs has been shown to aid in the identification of effector genes as well as to identify the causal variants for complex traits [45]. To this end, we generated 95% credible sets of sQTLs and eQTLs. For sQTLs, the highest credible set posterior probabilities (CPP) of fine-mapped variants were observed for markers in 5′ and 3′ splices sites, followed by exonic and intronic regions (Mann-Whitney $p = 1.36 \times 10^{-19}$, $1.13 \times 10^{-21}$, $4.55 \times 10^{-80}$, and $1 \times 10^{-3}$, respectively, compared with intergenic variants) (Fig. 4a). Seqweaver, which provides variant effect predictions using RBP-based deep learning models [46], showed increased disease impact scores (DIS) of sQTLs in analogous annotations (Additional file 1: Fig. S9a). Furthermore, fine-mapped intronic and exonic sQTLs disrupted motifs of auxiliary splicing regulators (*SRSF3*, *SRSF9*, *HNRNPA1*, *HNRNPC),* while those near 3′ splice sites showed recurrent disruption of branch point motifs and core splicing components [47, 48] (Additional file 1: Fig. S9c). These orthogonal analyses were consistent with known determinants of RNA splicing and highlight the potential of our fine-mapped sQTLs to prioritize causal variants.

Analogously, genetic fine-mapping of eQTLs revealed higher CPPs for variants in islet active promoters and enhancers (Mann-Whitney $p = 3.24 \times 10^{-203}$, $1.38 \times 10^{-58}$, in

promoters and mediator-enriched enhancers, respectively) (Fig. 4b). Similar enrichments were obtained with DeepSEA [49] (Additional file 1: Fig. S9b), and for disruption islet TF motifs [50–52] (Additional file 1: Fig. S9d). These results again supported that fine-mapped QTL variants have increased likelihood of driving splicing and expression variation in human islets.

Next, we hypothesized that if fine-mapped QTL variants are truly enriched in causal T2D variants, they should converge with variants that have highest CPPs in GWAS fine-mapping studies. To investigate this, we examined 99% credible set variants from T2D GWAS signals [1] with colocalizing QTLs (PP4>0.8, 16 loci for splicing, 28 loci for expression). We observed that GWAS credible set variants that were also in sQTL credible sets had higher GWAS CPP than GWAS credible variants that were not in the sQTL credible sets (Mann-Whitney $p= 1.71 \times 10^{-32}$) (Fig. 4c). Likewise, GWAS credible set variants that were also in eQTL credible sets showed higher GWAS CPP than other GWAS credible set variants (Mann-Whitney $p= 1.16 \times 10^{-62}$) (Fig. 4d). This convergence between GWAS and QTL fine-mapping provided further evidence that QTLs are likely to contain causal T2D risk variants. The integration of QTL and GWAS credible sets increased the number of associated loci with $\leq 5$ putative causal variants from one to five loci by integrating fine-mapped sQTLs, and 8 to 13 loci with fine-mapped eQTLs (Fig. 4e, f; see also examples in Fig. 4g and Additional file 1: Fig. S9e). Thus, fine mapping QTLs has the potential to boost the genetic resolution of GWAS credible sets.

*ERO1B* represents an example in which we fine-mapped a putative causal variant for a candidate effector gene. *ERO1B* showed a significant sTWAS T2D association in our studies ($p= 6.5 \times 10^{-6}$, significant at FDR $\leq 1\%$). Fine-mapping highlighted an exonic rs2477599 variant that localizes to a splicing silencer *HNRNPC* motif, which causes an exon skipping event that results in premature truncation of the *ERO1B* open reading frame (Fig. 4g). Previous genetic loss-of-function studies have shown that *ERO1B* (endoplasmic reticulum oxidoreductase 1 beta, also known as *ERO1LB*) is an ER protein that plays a critical role in insulin biosynthesis and β-cell survival [53–55]. rs2477599 has not been previously reported in T2D GWAS studies, but shows suggestive associations in GWAS for T2D ($p= 8.2 \times 10^{-6}$ in [1], and $p= 3.42 \times 10^{-5}$ in FinnGen Biobank, data freeze 7) and random glucose ($p= 8.3 \times 10^{-5}$ UK Biobank Mendelian traits [56]). These findings point to a fine-mapped putative causal splicing variant and a plausible effector mechanism for T2D susceptibility.

sQTLs also shed interesting findings at the *BCAR1*/*CTRB2* locus. GWAS have fine-mapped an intergenic variant near *CTRB2* (rs72802342, CPP=0.66) that is associated with T2D [1], T1D [5], and pancreatic ductal adenocarcinoma (PDAC) [57] and co-localizes with an accessible chromatin region in human acinar cells and islets [5, 58]. *CTRB2* showed significant sTWAS associations with T2D and T1D (Additional file 9: Table S8), and our fine-mapped sQTLs pointed to the same lead candidate causal variant, rs72802342, which appeared to cause a complete exon-skipping event in CTRB2 transcripts (Additional file 1: Fig. S9e). Recent work, however, has shown that rs72802342 tags a 584 bp deletion that completely overlaps exon 6, and explains our observed sQTL [57]. Thus, an exon-skipping deletion at *CTRB2* (rather than an exon skipping splice variant) is likely to be the functional causal variant underlying T2D, as well as T1D and PDAC, susceptibility.

### Islet QTLs provide insights into T2D pathways

The availability of an expanded list of candidate effector genes, as opposed to lists of genes that are located in the vicinity of associated SNPs, allowed us to explore the hypothesis that a subset of genetic signals could influence T2D susceptibility by acting on specific cellular pathways in pancreatic islet cells. We thus compiled 106 putative effector genes for T2D or glycemic traits (FG, FI) reported here as well as previously reported co-localizing islet eQTLs [12]. This list excluded those exclusively detected in non-endocrine cells after analysis of single cell RNA-seq datasets (Additional file 13: Table S12), as well as non-coding transcripts and genes without known function. Functional gene annotations revealed notable enriched pathways, including regulators of fatty acid biosynthesis (*SCD5*, *FADS1*, *GCDH*, *BDH2*, *PAM*; GO:0006636, GO:1901570, ENRICHR adjusted $p = 0.04$), and genes upregulated by hypoxia or mTORC1 activation (MsigDB Hallmark Hypoxia and mTORC1 signaling; $q < 0.05$ and $< 0.001$, respectively). We further used the list of 106 genes to identify networks using STRING [59] v11.5, using co-expression, experimental, and functional annotation databases. We allowed for minimal network inflation (≤5 interactions) and omitted text-mining to preclude bias arising from publications that name genes at significant GWAS loci. The resulting network exhibited 1.8-fold greater protein-protein interactions than random sets (PPI enrichment $p < 10^{-4}$) and contained two distinct subnetworks (Fig. 5a). One subnetwork contained components of the eIF3 translational initiation complex (FDR = 0.018). Several monogenic diabetes genes target translational initiation, in particular eIF2 complex (*EIF2B1*, *EIF2S3*, *EIF2AK3*), due to their impact on endoplasmic reticulum stress in β-cells [60, 61] (Fig. 5a). This process is also regulated by ERO1B [55]. Another notable sub-network was formed by molecular mediators of G-protein mediated enhancement of cAMP, a major insulinotropic pathway [62, 63]. Manual curation revealed eight well-characterized genes in this pathway, including *GLP1R*, *RGS17*, *PDE8B*, four members of the GNAI3 (G(i) subunit α3) protein-protein interaction complex (*GPSM1*, *RGS19*, *ADRA2A*, *ADCY5*), and *VSNL1*, a modulator of cAMP and insulin secretion [34] (Fig. 5a, b).

### Discussion

This study adds splicing variation in pancreatic islets to the spectrum of molecular mechanisms that underlie T2D predisposition. Earlier studies had examined expression QTLs in human pancreatic islets [10–13, 18, 19]. The current study offers the first systematic analysis of how common genetic variants influence RNA splicing in human islets. Parallel profiling of splicing and expression QTLs in the same dataset demonstrated that these represent two distinct mechanisms through which genetic variation can influence islet biology and disease. We found islet sQTLs that impact genes that have major roles in islet cell function as well as in the pathophysiology or treatment of various forms of diabetes. Furthermore, we observed a selective inflation of T2D association p-values among sQTLs, some of which showed fine co-localization with T2D variants. Finally, we applied for the first time splicing and expression TWAS to nominate T2D target genes and identified novel T2D genetic association signals.

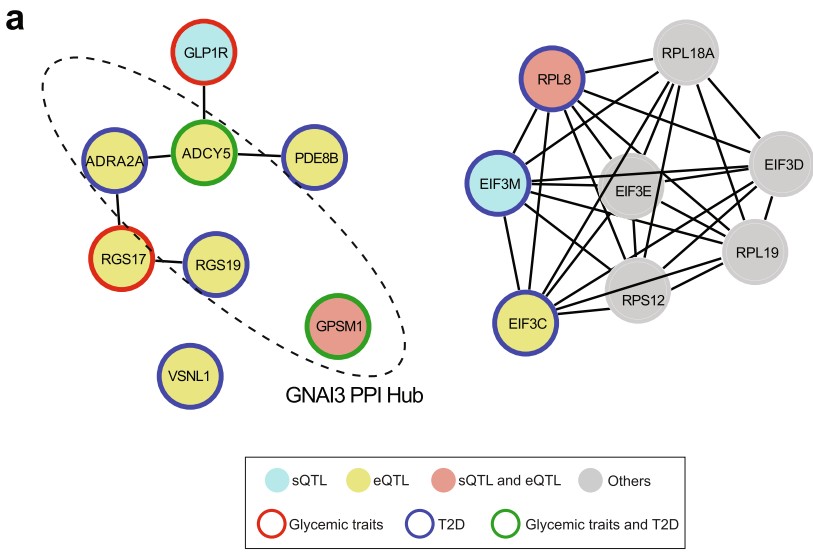

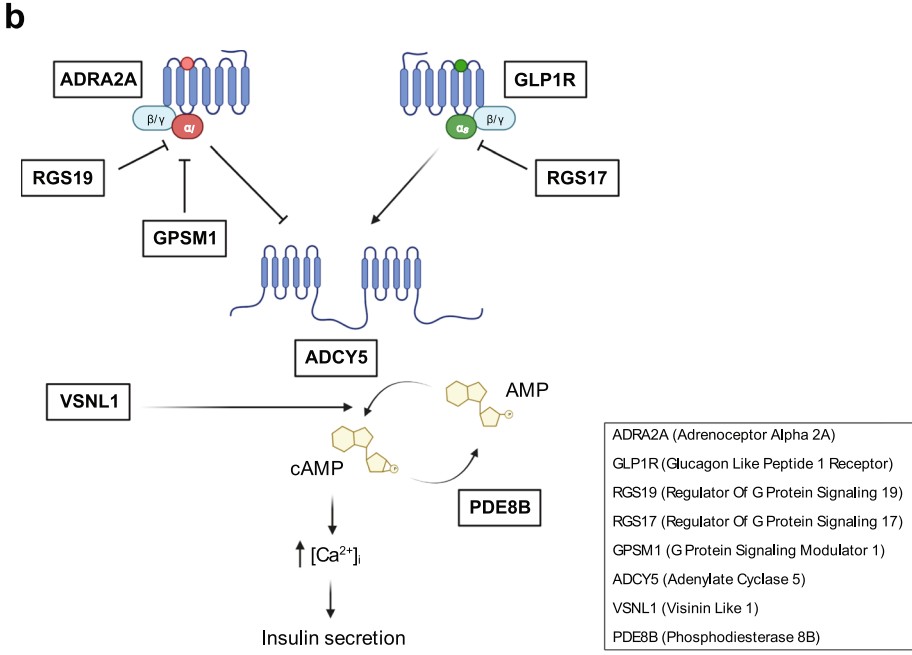

**Fig. 5** Islet QTLs that co-localize with T2D signals target distinct pathways. **a** STRING v11.5 was used to analyze 106 genes with sQTLs or eQTLs co-localizing with T2D or glycemic traits, including previously reported eQTLs. We only considered genes with known or presumed protein-coding function, and not exclusively expressed in non-endocrine cells according to single cell RNA-seq. We allowed for inflation of ≤ 5 interactors, and used default (>0.4) confidence scores. Shown are two networks with >3 components, one populated by components of heterotrimeric G protein signaling, and another by genes involved in eIF3 translational initiation. Genes are colored as indicated in the legend; other interactors added through STRING analysis are shaded gray. **b** Manual curation was used to illustrate the relationship between components of the G protein-mediated insulinotropic pathway targeted by islet QTLs linked to T2D and related traits

Gene targets that are nominated based on human genetic evidence double their likelihood of success in drug development pipelines [64]. We have expanded the current list of putative effector genes that mediate diabetes risk at known and novel susceptibility regions, and include examples that provide insights into T2D pathophysiology.

This was illustrated by the association of T2D risk with a fine-mapped splice variant that creates a premature stop codon in *ERO1B*. This newly reported locus is supported by an independent T2D association signal observed in FinnGen, as well as prior experimental and genetic studies that point to a central role in ER homeostasis, insulin biosynthesis, and diabetes progression [53–55]. Our analysis of sQTLs revealed a spurious exon-skipping splicing event in *CTRB2* caused by a deletion. This exon deletion, which has been previously characterized in the context of PDAC susceptibility [57], provides a more compelling candidate molecular driver for T2D and T1D susceptibility at this locus than recently reported transcriptional variants [5, 58] and can be tested in models to determine pathogenic mechanisms.

T2D has a highly polygenic architecture, with a very large number of effector genes that individually exert small effects. It remains possible, however, that many such genes converge on a small number of biological pathways. The extended list of genes with islet QTLs that co-localize with GWAS signals allowed us to explore shared pathways underlying T2D susceptibility. We found supportive evidence for a role in translation, ER stress, and fatty acid metabolism, and, most clearly, eight out of 106 QTLs that co-localized with glycemic trait or T2D associations were linked to G protein-coupled receptor (GPCR) signaling through cAMP. This pathway has been extensively involved in transducing signals from a vast range of extracellular insulinotropic stimuli, including incretin hormones, neurotransmitters, nutrients such as fatty acids, or extracellular matrix components [62, 65, 66]. Effectors from this pathway included an sQTL for *GLP1R*, a major drug target for T2D [67], four members of the same protein interaction complex, and a novel T2D association for *VSNL1*, previously shown to stimulate cAMP production and insulin secretion [34]. These findings, therefore, suggest that abnormal production of cAMP plays a causal role in T2D susceptibility, plausibly through its impact on insulin secretion and islet gene transcription. This pathway is a known therapeutic target to stimulate insulin secretion. However, our findings raise the additional prospect that it is possible to target altered GPCR signaling in precision medicine strategies that aim to correct causal defects, and thereby modify disease progression in susceptible individuals.

Our resource of islet sQTLs is relevant for efforts to dissect multi-allele interactions. There is growing evidence that disease-associated haplotypes can contain more than one functionally interacting causal variant [68]. Islet splicing variants can thus act in *cis* with other functional variants to influence disease susceptibility. For example, sQTLs could modify the penetrance of causal coding or cis-acting variants in both polygenic or Mendelian settings. Likewise, we have shown that islet sQTLs can alter drug target genes and can thus be examined to understand how genetic variation alters the response to therapies.

Finally, this molecular QTL resource holds relevance for the biological interpretation of genetic variants that influence autoimmune T1D risk, in particular for efforts to link biological processes in β-cells into T1D predisposition. Of interest, our findings provide a genetic finding that aligns with recent proposals that intrinsic β-cell senescence processes contribute to autoimmune β-cell destruction [42, 43].

Our analysis of molecular QTLs entails some limitations. Splicing activity was estimated by junction usage, which does not directly inform isoform level regulation, and therefore provides a partial picture of how local splicing translates into gene function.

Furthermore, the analysis of molecular QTLs was limited to islets, where genetic variants are known to play a central role in diabetes mellitus, most clearly in T2D. Some disease associations are expected to be mediated through other cell types and the mediating mechanism is not captured by our studies. In addition, our study was performed on bulk pancreatic islet samples with heterogeneous cell-type compositions, after exposure to conventional culture conditions. This limits the power to discover new QTL signals that are confined to minor cell types, or conditioned by specific environmental contexts [69]. Furthermore, colocalization and TWAS evidence does not preclude horizontal pleiotropy, and we have thus not unequivocally linked non-coding variant disease associations with the causal molecular targets. Nevertheless, we have demonstrated genetic co-localization of disease risk variants and molecular variation in pancreatic islets, which play a central causal role in diabetes. This knowledge can thus be used to nominate plausible effector mechanisms and should warrant further orthogonal genetic and experimental studies to establish causality.

## Conclusions

Our findings expose widespread effects of common genetic variants on RNA splicing in pancreatic islets, a tissue that is directly relevant to the pathophysiology of T1D and T2D. We highlight examples in which genetic and biological evidence supports a role in diabetes susceptibility. The results provide new avenues and resources to understand the genetic underpinnings of diabetes mellitus.

## Methods

### Human pancreatic islet datasets

We compiled 447 human islets samples, which comprised 89 samples from GEO accession number GSE50244 [19], 118 from EGA accession number EGAD00001001601 [18], 112 samples from T2DSystems EGA accession number EGAS00001005535 [13], and 128 samples from CRG cohort (EGAS00001006440) [70]. Out of 447 samples, 399 samples passed genotype and RNA-seq QC filters described below.

For the CRG cohort, human pancreatic islets from organ donors without a history of glucose intolerance were purified using established isolation procedures [71–74], shipped in culture medium and re-cultured at 37 °C in a humidified chamber with 5% $CO_2$ in glucose-free RPMI 1640 supplemented with 10% fetal calf serum, 100 U ml$^{-1}$ penicillin, 100 U ml$^{-1}$ streptomycin, and 11 mM glucose for 3 days before analysis. Islet isolation centers had permission to use islets for scientific research if they were insufficient for clinical transplantation following national regulations and ethical requirements and institutional approvals from University of Lille, University of Geneva, and Milano San Raffaele Hospital. Ethical approval for processing de-identified samples was granted by the Clinical Research Ethics Committee of Hospital Clinic de Barcelona and Parc de Salut Mar.

### Genotype processing

The 128 CRG cohort samples were genotyped with Illumina Infinium OmniExpress 12 v1 and HumanOmni 2.5-8v1 on a total of 624K SNPs. The 89 samples from Fadista et al. (GSE50244) [19] were genotyped using Illumina HumanOmniExpress 12v1 C on

a total of 609K SNPs. The 118 samples from Van de Bunt et al. (EGAD00001001601) [18] were genotyped using Illumina Omni2.5+Exome array on a total of 2.6M SNPs. Finally, the 112 samples from T2DSystems (EGAS00001005535) [13] were genotyped using Illumina's Human Omni 2.5 exome array on a total of 2.6M SNPs.

A three-step quality control of genotype data, involving two stages of SNP removal and one intermediate stage of sample exclusion, was conducted in each cohort. Genotyped SNPs were filtered if (i) minor allele frequency (MAF) < 0.01, (ii) missing genotype rate $\geq$ 5%, and (iii) significantly deviated from Hardy-Weinberg equilibrium (HWE $p$-value $\leq 1\times10^{-6}$). Samples were excluded if (i) individual missing genotype rate $\geq$ 2%, (ii) cryptic relationships and sample duplicates (individuals with higher individual missingness genotype rate from pairs with pi $\geq$ 0.185), or (iii) showed >4 standard deviations from the mean according to the first four principal components in each given cohort.

After QC analysis for genotype data, (i) 113 individuals and 557,422 SNPs were retained for the CRG cohort, (ii) 112 individuals and 1,499,688 SNPs for the Van de Bunt et al. cohort [18], (iii) 89 individuals and 596,464 SNPs for the Fadista et al. [19] cohort, and (iv) 109 individuals and 1,543,968 SNPs for the T2DSystems cohort [13].

For each cohort, we generated per-chromosome VCF files after removing all strand ambiguous variants (AT or CG SNPs), and checking for strand alignment against the Haplotype Reference Consortium (HRC) and 1000 Genomes (1000G) reference SNP list. *HRC-1000G-check-bim.pl* script with the *-n* option (to turn off the removal of variants showing MAF differences between the reference panel and the study genotypes) was used. We submitted resulting VCF files to the Michigan Imputation Server (https://imputationserver.sph.umich.edu/index.html): EAGLE2 [75] was used for phasing, minimac3 [76] for genotype imputation with the HRC [77] r1.1 and the 1000G Phase 3 release reference panels [78], independently. For each dataset of imputed genotypes, we excluded variants with (i) MAF < 1%, (ii) imputation-quality $R^2$ < 0.7, and/or (iii) HWE $p$-value $\leq 1\times10^{-6}$. We extracted indels from the 1000G Phase3 imputed results, filtered them using the aforementioned criteria, and merged with the filtered HRC imputed dataset.

### RNA-Seq data alignment and QC

Raw fastq files from all cohorts were aligned to the hg19 genome build using STAR [79] and the options, *--outFilterMultimapNmax 1 --outSAMstrandField intronMotif --outSAMattributes All --twopassMode Basic*. WASP [80] pipeline was used to remove reads mapped with allelic bias. *VerifyBAMID* [81] was used to assess the concordance between genotypes and RNA-Seq, and samples with more than 2% contamination (CHIPMIX > 0.02 and FREEMIX > 0.02) were removed. This resulted in 101 samples from CRG cohort, 109 from samples Van de Bunt et al. cohort [18], 82 samples from Fadista et al. [19] cohort, and 107 samples from T2DSystems [13]. The resulting BAM files were used for gene expression quantification and to calculate splicing activity. In-case of EGAD00001001601 [18], only bam files were available; hence, the initial STAR alignment step was not performed.

### Principal component analysis of genotypes

To identify individuals of divergent ancestry and to characterize population structure of 399 human islet samples, we first selected a subset of genotyped SNPs that were common in all 4 cohorts, that also passed all our QC filters (see the "Genotype processing" section), and with MAF $\geq$ 1% and missingness < 5% across all the samples. We also excluded SNPs in high LD (pairwise $r^2 \leq 0.1$ within 1 Mb window), C/G and A/T SNPs to avoid strand mismatches, and those located in previously reported regions with long-range LD. We aggregated the 1000 Genomes Phase3 reference dataset using the set of overlapping variants. flashPCA [82] tool was used to calculate genetic principal components (PCs) (Additional file 1: Fig. S1c-d). First four genetic PCs are used as covariates in subsequent sQTL and eQTL mapping analysis.

### Gene expression quantification

In-house developed human pancreatic islet transcriptome annotations built with a combination of CAGE as well as short and long sequence reads were used to quantify total gene expression (Atla, G., Beucher, A. et al., unpublished). FeatureCounts [83] was used to obtain gene level qualifications using default parameters except using appropriate strandedness flag for each dataset. Genes with less than 5 raw reads mapped in less than 10% of the samples within each cohort were removed. CPM normalization was performed using edgeR [84] *cpm* function and then log2 transformed. Combat [85] was used to remove known sequencing batch effects. Principal components (PCs) were calculated on Combat corrected gene expression values using *prcomp* function in R (Additional file 1: Fig. S1a).

### Splicing quantification

To quantify splicing activity, we used the annotation free method, *leafcutter* [20]. Briefly *bam2junc.sh* script from leafcutter was used to quantify junction spanning reads based on spliced alignments from bam files. We removed junctions that are not supported by at least 5 spliced reads in at least 10% of samples. We then used *leafcutter_cluster.py* script with options *-m 30 -l 500000* to cluster the junctions anchoring on shared splice sites. This identifies local splicing events which are a cluster of alternatively used junctions. *prepare_phenotype_table.py* script was used to get relative junction usage (ratios) across samples. The relative junction usage is also referred to as percent spliced in (junction PSI) in the manuscript. Combat [85] was used to remove known sequencing batch effects and principal components were generated using *prcomp* function in R (Additional file 1: Fig. S1b).

### cis-eQTL mapping

*cis*-eQTL mapping was performed using QTLtools [86] for 399 samples with available QCed genotype and RNA-seq data using a cis-window of 500 kb up- and downstream of the transcription start site (TSS). Fifteen PCs derived from gene expression and 4 genetic PCs were used as covariates in the linear model. In order to identify best associated cis eQTL SNP-eGene pairs, QTLtools was run using the permutation pass mode using default parameters and *--permute 1000 --window 500000 --seed 123456.*

Beta approximated permutation *p*-values were adjusted for multiple testing correction using Storey *q*-values implemented in the *qvalue* R package. We set the significance threshold at FDR *q*-value ≤ 0.01. This resulted in 3433 genes (eGenes) with significant eQTLs (Additional file 2: Table S1). We also calculated nominal *p*-values for all cis-SNPs within a 500-kb window centered on the TSS of each gene (nominal pass mode from QTLtools, *--nominal 1 --window 500000 --seed 123456*). To identify all significant variant-gene pairs, we defined a genome-wide *p*-value threshold (*pt*), by considering the empirical *p*-value of the eGene closest to the 0.01 FDR threshold. A gene-based nominal *p*-value threshold was then calculated using *pt* and the beta distribution parameters from QTLtools. For each significant eGene, variants with a nominal *p*-value below the gene-level threshold were considered in subsequent analyses (significant nominal cis-eQTL variants) [21].

### cis-sQTL mapping

We performed cis-sQTL mapping as described for cis-eQTL identification using intron excision ratios and a cis-window of 50 kb up- and downstream of the junction (*--window 50000*). In case of cis-sQTLs, 5 PCs derived from splicing ratios and 4 genetic PCs were used in the linear model. In order to identify best associated cis sQTL SNP-junction pairs, QTLtools was run using the permutation pass mode (1000 permutations), and beta approximated permutation *p*-values were adjusted for multiple test correction using Storey *q*-values implemented in the *qvalue* R package. We set the significance threshold at FDR *q*-value ≤ 0.01, resulting in 4858 junctions with a significant sQTL (Additional file 3: Table S2). We also calculated nominal *p*-values for all cis-SNPs within a 50-kb window around the junction (nominal pass mode from QTLtools). To identify all significant variant-junction pairs, we defined a genome-wide *p*-value threshold (*pt*) by considering the empirical *p*-value of the junctions closest to the 0.01 FDR threshold. A junction-based nominal *p*-value threshold was then calculated using *pt* and the beta distribution parameters from QTLtools. For each significant junction, variants with a nominal *p*-value below the junction-level threshold were considered in subsequent analyses (significant nominal cis-sQTL variants) [21].

### Imputation information score of lead eQTLs and sQTLs

We used QCTOOL v2.0.6 to calculate imputation scores (*-snp-stats)* for each lead eQTL and sQTL identified in 3433 genes and 4858 splicing junctions at FDR ≤ 1%, respectively.

### Annotation of sQTL junctions

As leafcutter identifies junctions de-novo, we used transcriptome annotation GTF files as a base to annotate the junctions with respective genes. We used *gtf2leafcutter.pl* script from leafcutter's leafviz module (https://github.com/davidaknowles/leafcutter/tree/master/leafviz) to obtain the intron coordinates of all genes. The sQTL junctions were then mapped to the intron coordinates and annotated with respective gene names.

### Magnitude of genetic effects on splicing

To quantify the magnitude of genetic effects on splicing, for each cluster, we chose a junction with the best *q*-value. Then, for each sQTL junction, we calculated the

difference in median junction usage (delta-psi) from samples with homozygous reference and homozygous alternate alleles and plotted as a function of the $-\log_{10}$ ($q$-value). If alternate homozygous samples were not available, we chose median junction usage from heterozygous samples.

### Visualization of splicing events

The junctions identified from leafcutter were loaded into IGV [87] for visualization as arcs, and box plots were plotted using python.

### Comparison of eQTLs and sQTLs

Our maps of genetic effects on splicing were compared to our eQTLs but also to previously reported exonQTLs and eQTLs in the largest study to date in human pancreatic islets by the InsPIRE consortium [12]. To estimate the gain of novel information uniquely provided by our sQTL analysis, we first determined the overlap between sGenes and eGenes in our study, and eQTLs and exon QTLs mapped by InsPIRE. For those genes that contain both sQTLs and either eQTLs or e/exon QTLs in InsPIRE, we calculated the LD ($r^2$ measure) between the lead sQTL and all other SNPs within 1Mb using PLINK [88] (v1.9 *--ld-window-kb 1000 --ld-window 99999 --ld-window-r2 0*). As a reference dataset for LD calculations, we used our high-quality genotypes in 399 samples. Then, for each gene, we plot the LD $r^2$ distribution between the lead sQTL and the lead QTL from the dataset being compared, respectively.

### Genomic enrichment analysis of eQTLs and sQTLs

We used GREGOR [89] to perform enrichment analysis of lead sQTLs and eQTLs in different genomic annotations using the following options R2THRESHOLD=0.7, LDWINDOWSIZE=50000 (for sQTLs), LDWINDOWSIZE=100000 (for eQTLs), MIN_NEIGHBOR_NUM=200, and POPULATION=EUR.

Human islet regulatory annotations were described previously [11]. For genic annotations, we used Gencode [22] v34 GTF file. We used *gtf2leafcutter.pl* script to obtain all exon, intron, 5′ and 3′ coordinates from GTF file. 5′ and 3′ coordinates were extended to +3bp into exon and +2bp into intron. All annotations contain mutually exclusive genomic space.

In parallel, we used GARFIELD [90] for enrichment analysis among genomic annotations. We used QTLtools beta approximated permutation $p$-values of lead eQTL and sQTLs from all genes and junctions tested in cis-eQTL and sQTL analysis. If a variant was lead eQTL or sQTL for multiple genes or junctions, we kept the lowest $p$-value. We used the same annotations as in GREGOR. For each set of eQTLs and sQTLs, the method first removes variants with $r^2 \geq 0.1$ in a 1-Mb window from the most significantly associated variant, and then links each of the independent variants to genomic annotations if either the variant or a variant in LD ($r^2 \geq 0.8$) overlaps a given feature. Finally, the method tests for enrichment with a logistic regression model that controls for confounding factors such as distance to the TSS, number of LD proxies, and MAF that we binned in 4 quantiles. We used three different QTL $p$-value thresholds to test for enrichment, $5 \times 10^{-3}$, $5 \times 10^{-5}$, and $5 \times 10^{-7}$. Significant enrichments for eQTLs and sQTLs were identified after applying multiple-test correction for the effective number of

annotations (estimated by GARFIELD), the number of QTL *p*-value thresholds, and the number of molecular QTL phenotypes.

### Comparison with GTEx

We obtained summary statistics data for eQTLs and sQTLs from 49 GTEx tissues [17]. We first obtained eGenes and junctions with significant eQTLs and sQTLs, respectively, at FDR 0.01, consistent with our significance threshold. For the resulting significant variant-phenotype associations for each of the 49 tissues, variant position and junction coordinates (for sQTLs) were lifted down from hg38 to hg19 using liftOver [91]. For each GTEx tissue, we looked at the variant-phenotype (eGenes or Junctions) overlap with our islet eQTL and sQTLs, using nominal QTL variants. For example, for GTEx *x* eGene in *j* tissue, if any of the GTEx significant variants mapped any of our nominal eQTL variants for that *x* eGene, we considered that islet eQTL signal to be shared with that given *j* tissue. Same approach was implemented to sQTLs. We excluded from this analysis testis, given the pervasive number of eQTLs, and pancreas because it is a partial surrogate of pancreatic islets.

### Quantile-quantile plots

We generated quantile-quantile (Q-Q) plots to estimate genomic inflation in sQTLs and eQTLs for T2D and T1D risk, and glycemic traits variation. For T2D, we used BMI-adjusted T2D summary statistics [1]. For glycemic traits, we leveraged summary statistics data from a trans-ancestral meta-analysis for fasting glucose (FG) and fasting insulin (FI) [23]. For T1D, we used summary statistics data from a recent large-scale meta-analysis [5]. We included variants with MAF $\geq$ 5% that were intersected with our nominal e- and sQTLs. For each trait comparison, we generated 1000 permutations of subsets of control sQTL variants to provide further support to the observed enrichment of e- and s-QTLs among GWAS variants. Each control set of sQTL-like variants was generated by first identifying independent LD blocks [92], that comprised nominal eQTL or sQTL variants, respectively. Then, we shuffled non-overlapping genomic regions that were created based on the size of the genomic ranges where our nominal sQTL variants were located, across the genome, but excluding those independent LD regions where either eQTL or sQTL variants were located, blacklisted regions (wgEncodeDacMapabilityConsensusExcludable.bed.gz and wgEncodeDukeMapabilityRegionsExcludable.bed.gz) and the MHC region. Among the set of shuffled independent LD regions, we randomly sampled the same number of nominal sQTL variants. This was done 1000 times.

For the specific purpose of addressing T2D susceptibility across shared or independent sQTL/eQTL effects, we considered sQTLs and eQTLs from junctions and eGenes whose corresponding lead sQTL and eQTL were in $r^2 \geq 0.6$ as shared. The rest were considered as sQTL or eQTL specific genetic effects, respectively. To estimate T2D risk inflation across islet-selective sQTLs and eQTLs, we grouped sQTLs and eQTLs that were shared in $\leq 6$ GTEx tissues (see the "Comparison with GTEx" section) as islet-selective QTL effects. To assess the strength of T2D-risk inflation, we leveraged 1000 permutations of control-sets of sQTL-like variants as described above.

**Colocalization analysis across T2D, T1D and FG/FI independent GWAS signals**

We collected 403 independent lead variants for T2D [1] and 277 for FG and FI [23] and 136 for T1D [5]. For each trait, we performed colocalization as implemented in *gwas-pw* [24] at each independent GWAS signal. We only tested for colocalization with e- and sQTLs at those signals with at least one credible set variant (genetic posterior probability $\geq 0.01$) in LD ($r^2 \geq 0.6$) with a lead eQTL or sQTL, respectively. Credible sets were not available for FG and FI GWAS. Thus, we tested for colocalization at FG/FI GWAS signals if at least one variant with GWAS $p \leq 5\times10^{-5}$ was in LD ($r^2 \geq 0.6$) with a given lead eQTL or sQTL. GWAS signal LD calculations were performed using the high-quality genotypes from our 399 islet donor samples. If for a given independent GWAS signal, any of the available proxy variants was not included in our imputed genotypes, we used 1000 Genomes Phase3 genotypes (with European descent) as the reference panel [78]. Colocalization was performed across 1Mb genomic interval centered on the reported lead independent variant for a given GWAS locus, and including variants with a GWAS $p \leq 5\times10^{-5}$. We nominated a region as a colocalized locus if the posterior probability for the model 3 (presence of the same genetic variant underlying the QTL and GWAS association, "colocalization") was $\geq 0.8$. Colocalization signals were visualized using R-3.6.1 and LocusCompareR 1.0.0.

**Visualization of regional association plots**

Regional association plots for T2D and T1D susceptibility regions were created using LocusZoom [93] v1.4 in R-3.6.1.

**TWAS analysis**

FUSION [25] was used for TWAS analysis with T2D, T1D, and glycemic traits (FG and FI). For gene expression, first the weights were computed using *FUSION.compute_weights.R* and options *--models top1, blup, bslmm, lasso, enet* on the same data that is used for eQTL analysis. Fifteen gene expression PCs and 4 genetic PCS were used as covariates and variants within a cis-window of 500kb from TSS were used. For splicing analysis, variants within 50kb from the junction boundaries were used and 5 splicing PCs and 4 genetic PCs were used to compute the weights.

After computing weights, *FUSION.assoc_test.R* script was used to test for association of the pre-computed weights and GWAS summary statistics data for T2D [1], T1D [5], and glycemic traits [23] (FG and FI). We only included GWAS data from variants that overlaid our ~6.5M high-quality imputed common genetic variants. Multiple test correction was applied to the resulting *p*-values using Bonferroni. We also excluded TWAS results with low colocalization posterior probabilities and with large confounding effects from linkage (gwas-pw PPA_3 or COLOC PP4 < 0.6).

We mapped TWAS results to T2D, glycemic, and T1D GWAS loci by identifying TWAS results whose best GWAS lead variant was in LD ($r^2 \geq 0.1$, calculated using the genotypes of individuals with European descent from the Phase 3 of the 1000 Genomes Project [78]) or less than 500kb away from lead GWAS signals, for each trait respectively. The rest of TWAS results were considered as novel GWAS loci. For T2D, we also considered as known GWAS loci if the TWAS signals are in LD ($r^2 \geq 0.1$) with lead independent signals identified in a recent large-scale meta-analysis [3].

### Conditional analysis in the PTPN22 T1D locus

We performed conditional analysis on the *DCLRE1B* sQTL using COJO [94], GWAS summary statistics from a recent T1D meta-analysis [5] and the genotypes from 1000 Genomes Phase 3 as the LD reference panel. We first conducted COJO single-variant analysis (*gcta --cojo-cond*) that conditioned on the lead *PTPN22* missense variant (rs2476601) and the *DCLRE1B* sQTL (rs11102694), respectively. Additional joint SNP analysis (*--cojo-joint*) was performed with COJO considering both *PTPN22* missense and the *DCLRE1B* sQTL variant. For the joint SNP analysis, we used a fraction of 381,380 unrelated UK Biobank individuals as the LD reference panel (subjects related up to second-degree were filtered) of white European origin.

### Single cell gene expression analysis

We quantified published single cell data sets [95–98] against in-house annotations (Atla, G., Beucher, A., et al., unpublished) using featureCounts [83]. Seurat V3 [99] was used to normalize and identify cell-types within each data set. A gene is annotated as expressed if it has normalized counts of $\geq 0.3$ in at least 30 cells of a given cell-type in one of the data sets.

### Network analysis

We selected all the nominated effector genes for T2D, FG, FI from the current study and InsPIRE [12], filtered for 106 that encode for proteins with known or presumed function, and used them to construct networks using StringDB [100]. StringApp [101] in cytoscape [102] was used to identify protein-protein interaction networks using default parameters, confidence score cutoff 0.4 and maximum additional interactors of 5. This yielded 1.8-fold higher protein-protein interactions than random sets, although only subnetworks with 3 or more components are shown.

### Credible set analysis

We used fine-mapping approaches to identify candidate causal variants that underlie cis-eQTL and sQTL *loci* [21]. We identified 95% credible set variants using CAVIAR [103] software and allowing for one causal variant (*-c 1*). LD information between SNP pairs (i.e., the *r* matrix) was generated using PLINK [88] v1.9 *–matrix –r*, and as reference panel, we used the 399 human islet samples used in the eQTL and sQTL identification.

For each e/sQTL credible set, posterior probabilities were plotted with respect to underlying genomic annotations.

### In silico functional scores

The potential impact on disease was assessed for credible set variants of both eQTLs and sQTLs on the basis of their predicted transcriptional and post-transcriptional regulatory effects. We used a deep-learning model that is trained on transcriptional regulatory features such as histone marks, DNAse I profiles, and transcription factors, a total of 2002 features, and is implemented as DeepSEA [49] as well as post-transcriptional regulatory features such as RNA-binding proteins binding data based

on CLIP experiments on 82 unique RBPs (ENCODE and other CLIP datasets), which is implemented as Seqweaver [46]. We performed in silico mutagenesis on both eQTL and sQTL credible set variants using both DeepSEA and Seqweaver models and obtained Disease Impact Scores (DIS) values from each respective model [21]. Prior to in silico mutagenesis, the strand information was added to sQTL credible sets based on the orientation of the target gene. Credible sets were further stratified based on their location in the genome and DIS values from both models for each category of variants were shown as boxplots.

### Integrating GWAS credible sets with QTL credible sets

We obtained pre-calculated genetic credible sets for T2D [1] (https://diagram-consortium.org/downloads.html). For each colocalized locus (colocalization PP >0.8), we then annotated credible set variants into two mutually exclusive categories: (i) GWAS credible set variant which is also a QTL credible set variant and (ii) GWAS credible set variant which is not a QTL credible set variant but in LD ($r^2 < 0.1$) with a lead QTL. We then plotted the distributions of CPPs of all GWAS credible sets stratified into above two categories. If a variant belongs to more than one credible set, we used the maximum CPP of that variant.

### Supplementary Information

---

**Additional file 1: Supplementary Figures S1-S9 and figure legends**.

**Additional file 2: Table S1.** Significant lead eQTLs.

**Additional file 3: Table S2.** Significant lead sQTLs.

**Additional file 4: Table S3.** Examples of sGenes with known functions in islet biology or diabetes.

**Additional file 5: Table S4.** GREGOR enrichments of functional annotations in e/sQTLs.

**Additional file 6: Table S5.** GARFIELD enrichments of functional annotations in e/sQTLs.

**Additional file 7: Table S6.** sQTL colocalizations with gwas-pw (T2D/FG/T1D).

**Additional file 8: Table S7.** eQTL colocalizations with gwas-pw (T2D/FG/FI/T1D).

**Additional file 9: Table S8.** sTWAS (T2D, FG, FI, T1D).

**Additional file 10: Table S9.** eTWAS (T2D, FG, FI, T1D).

**Additional file 11: Table S10.** T2D loci with target genes nominated in this study and comparison with InsPIRE.

**Additional file 12: Table S11.** FG/FI loci with target genes nominated in this study and comparison with InsPIRE.

**Additional file 13: Table S12.** scRNA expression of nominated genes for all traits.

**Additional file 14: Review history.**

---

### Acknowledgements

We thank the Genomics Facilities from the Center of Genomic Regulation and NIHR Imperial Biomedical Research Centre. Members of the T2DSystems Consortium. Miriam Cnop: ULB Center for Diabetes Research, Medical Faculty, Université Libre De Bruxelles, Brussels, Belgium. Lena Eliasson: Lund University Diabetes Centre; Department of Clinical Sciences, Malmö, Lund University; and Clinical Research Centre, Skåne University Hospital, Malmö, Sweden. Jonathan Lou S. Esguerra: Lund University Diabetes Centre; Department of Clinical Sciences, Malmö, Lund University; and Clinical Research Centre, Skåne University Hospital, Malmö, Sweden. Décio L. Eizirik: ULB Center for Diabetes Research, Medical Faculty, Université Libre De Bruxelles, Brussels, Belgium. Anna L. Gloyn: Oxford Centre for Diabetes, Endocrinology and Metabolism, Radcliffe Department of Medicine, University of Oxford, Oxford, UK. Leif Groop: Diabetes Centre, Lund University, Malmö, Sweden. Thomas S. Jensen: Intomics A/S, Kgs. Lyngby, Denmark. Torben Hansen: Section of Metabolic Genetics, Novo Nordisk Center for Basic Metabolic Research, Faculty of Health and Medical Science, University of Copenhagen, Copenhagen, Denmark. Piero Marchetti: Department of Clinical and Experimental Medicine, and AOUP Cisanello University Hospital, University of Pisa, Pisa, Italy. Josep M. Mercader: Program in Medical and Population Genetics, Broad Institute of MIT and Harvard, Cambridge, MA, USA. Hindrik Mulder: Unit of Molecular Metabolism, Lund University Diabetes Centre, Malmö, Sweden. Rashmi B. Prasad: Lund University Diabetes Centre, Department of Clinical Sciences, Lund University, Skåne University Hospital, Malmö, Sweden. Chris R. Stabile-Barnett: A2F-Associates Limited, Suffolk, United

Kingdom. Christian Thirion: Sirion Biotech, Planegg-Martinsried, Germany. David Torrents: Barcelona Supercomputing Center (BSC), Barcelona, Spain. Jorge Ferrer: Centre for Genomic Regulation, The Barcelona Institute of Science and Technology, Barcelona, Spain.

### Peer review information

### Review history

The review history is available as Additional file 14.

### Authors' contributions

G.A., S.B.G., and J.F. conceived, coordinated the study, interpreted the results, and wrote the manuscript with input from remaining authors. A.B. analyzed the transcript annotations. I.M. provided guidance on transcriptome analysis during initial phases. M.C.A provided insights into pathway analysis and M.I. on RNA splicing. A.L.G and L.G. provided the datasets. T.B., F.P, L.M., J.K-C., M.S., P.M, and L.P. procured human islets. J.G-H. processed the RNA and DNA samples. D.J.M.C. and J.A.T. performed the conditional analysis. G.A. and S.B.G. processed the genome data and designed and performed all genetic and statistical analysis. All authors read and approved the final manuscript.

### Funding

This research was supported by Ministerio de Ciencia e Innovación (BFU2014-54284-R, RTI2018-095666-B-I00), Medical Research Council (MR/L02036X/1), a Wellcome Trust Senior Investigator Award (WT101033), European Research Council Advanced Grant (789055), EU Horizon 2020 TDSystems (667191), ESPACE (874710), and Marie Sklodowska-Curie (643062, ZENCODE). S.B.G was supported by a Juan de la Cierva postdoctoral fellowship (MINECO; FJCI-2017-32090). M.C.A was supported by a Boehringer Ingelheim Fonds PhD fellowship. Work in CRG was supported by the CERCA Programme, Generalitat de Catalunya, Centro de Excelencia Severo Ochoa (CEX2020-001049), and support of the Spanish Ministry of Science and Innovation to the EMBL partnership. Work in Imperial College was supported by NIHR Imperial Biomedical Research Centre. M.I. was supported by a European Research Council consolidator award (101002275). D.J.M.C. and J.A.T. were supported by JDRF grants 9-2011-253, 5-SRA-2015-130-A-N, 4- SRA-2017-473-A-N, and Wellcome grants 091157/Z/10/Z and 107212/Z/15/Z, to the Diabetes and Inflammation Laboratory, Oxford, as well as the Oxford Biomedical Research Computing (BMRC) facility, a joint development between the Wellcome Centre for Human Genetics and the Big Data Institute supported by Health Data Research UK and NIHR Oxford Biomedical Research Centre, and Wellcome Trust Core Award grant 203141/Z/16/Z. D.M.J.C analysis with the UK Biobank Resource was conducted under Application 31295. A.L.G. is a Wellcome Senior Fellow in Basic Biomedical Science and was supported by the Wellcome Trust (095101, 200837, 106130, 203141), the NIDDK (U01DK105535 and UM1 DK126185), and the Oxford NIHR Biomedical Research Centre.

### Availability of data and materials

All processed data files for the analyses described here, including all nominal sQTL and eQTL results, and fine-mapped eQTL and sQTL data, are available at Zenodo DOI: 10.5281/zenodo.6546807 [21].
Raw RNA-seq and genotyped SNP array data are available from the European Genome–phenome Archive (EGAS00001006440) [70].

## Declarations

### Ethics approval and consent to participate

Islet isolation centers had permission to use islets from donors for scientific research if they were insufficient for clinical transplantation following national regulations and ethical requirements and institutional approvals from Leiden University Medical Center, Geneva University Hospitals, University of Lille, and Milano San Raffaele Hospital. Ethical approval for processing de-identified samples was granted by the Clinical Research Ethics Committee of Hospital Clinic de Barcelona (HCB/2014/0926 and HCB/2014/1151) and Clinical Research Ethics Committee from Parc Salut Mar (CBS20_007A). All studies complied with the Helsinki Declaration.

### Consent for publication

Not applicable

### Competing interests

ALG's spouse is an employee of Genentech and holds stock options in Roche. JAT is a member of the GSK Human Genetics Advisory Board.

### Author details

[1]Centre for Genomic Regulation, The Barcelona Institute of Science and Technology, Barcelona, Spain. [2]Centro de Investigación Biomédica en red Diabetes y enfermedades metabólicas asociadas (CIBERDEM), Barcelona, Spain. [3]Department of Metabolism, Digestion and Reproduction, Imperial College London, London, UK. [4]JDRF/Wellcome Diabetes and Inflammation Laboratory, Wellcome Centre for Human Genetics, Nuffield Department of Medicine, NIHR Oxford Biomedical Research Centre, University of Oxford, Oxford, UK. [5]Present Address: Life Sciences Department, Barcelona Supercomputing Center (BSC), 08034 Barcelona, Spain. [6]Lund University Diabetes Centre, Clinical Research Center, Malmö, Sweden. [7]Department of Clinical Sciences in Malmö, Lund University, Malmö, Sweden. [8]Oxford Centre for Diabetes, Endocrinology and Metabolism, Radcliffe Department of Medicine, University of Oxford, Oxford, UK. [9]Department of Pediatrics, Division of Endocrinology, Stanford School of Medicine, Stanford, CA, USA. [10]Department of Clinical and Experimental Medicine, AOUP Cisanello University Hospital, University of Pisa, Pisa, Italy. [11]Cell Isolation and Transplantation Center, University of Geneva, Geneva, Switzerland. [12]Department of Medicine, Leiden University Medical

Center, Leiden, the Netherlands. [13]Hubrecht Institute/KNAW, Utrecht, the Netherlands. [14]University of Lille, Institut National de la Santé et de la Recherche Médicale (INSERM), Centre Hospitalier Universitaire de Lille (CHU Lille), Institute Pasteur Lille, U1190 -European Genomic Institute for Diabetes (EGID), F59000 Lille, France. [15]Diabetes Research Institute, IRCCS Ospedale San Raffaele and Università Vita-Salute San Raffaele, Milan, Italy.

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

## 
