## [**Additional file 14: Review history.** · Genome Biology]

Review History

First round of review

Reviewer 1

Are you able to assess all statistics in the manuscript, including the appropriateness of statistical tests used? Yes, and I have assessed the statistics in my report.

Comments to author:

Understanding the role of splicing in pancreatic islets with respect to conferring risk for type 2 diabetes and related traits is a key endeavor. The authors appropriately analyzed islets sQTLs in data derived from 399 donors. Given that GTEx does not currently include splicing data for pancreatic islets, this data represents a particularly important resource for variant-to-function efforts. Here are my comments:

1. In the Discussion, the authors state: "Splicing activity was estimated by junction usage, which does not directly inform isoform level regulation...". Given the authors are implicating a number of key events/genes, it would be optimal for a subset to be functionally validated/investigated with respect to isoform level - it would lend more credibility to the observations made.
2. Another limitation is that pancreatic islets are a mixed cell population and thus the splicing observed really represents a composite of what is seen across the cell types present - it won't be until such work is carried out in the single cell context that one would be able to assign a splicing event to a given cell type. And of course, splicing is a dynamic process so one only gets a snapshot at a given static time point. The authors are encouraged to mention these points in the Discussion.
3. The authors state that "Focusing on genes expressed in >10% of samples in each cohort..." was how they went about their analyses. This leads to a bias toward more abundant transcripts, and conversely could lead to the missing of potentially informative less abundant transcripts. The authors should describe why they selected this threshold.
4. On page 7, the word 'robust' is used in rapid succession. How is this defined in this particular context?

Reviewer 2

Are you able to assess all statistics in the manuscript, including the appropriateness of statistical tests used? Yes, and I have assessed the statistics in my report.

Comments to author:

Atla and colleagues have carried out a large expression and splicing QTL study in human pancreatic islets. They have obtained new associations, they have integrated these data with GWAS data for relevant diseases and traits and they have uncovered interesting new biological associations. This paper is of broad interest to others in the field.

While this study is properly conducted and advances new ground, I have a few comments, mostly methodological.

Main:

-Re fig 1 (and other parts of the ms): Enrichment was computed via enrichR/GREGOR did you try other methods to account for bias in background (e.g. FORGE2, GARFIELD)? I am concerned the p-values are a bit high and would like to see additional methods accounting for bias in enrichment.

-Re fig 3a: Grey random background set area seems truncated on the right and does not cover blue spot x axis position.

-"We thus compiled 106 putative effector genes for T2D or glycemic traits (FG, FI) reported here as well as previously reported co-localizing islet eQTLs¹⁰. This list excluded those exclusively detected in non-endocrine cells after analysis of single cell RNA-seq datasets (Supplementary Table 10), as well as non-coding transcripts and genes without known function." Why this extra non-endocrine filter?

Minor:

abstract: "This data exposes" (shouldn't this be "These data expose ")

Intro:

-"pointing to additional poorly understood genetic mechanisms." Is splicing poorly understood?

Main text:

-"This overlap increased to 90% in comparisons with unpublished human islet transcript annotations built with long-reads (GA, AB, unpublished) (Supplementary Figure 2b)." Can you add a preprint?

-"This indicates that sQTLs, which directly measure splice junction variation, and exon-QTLs, which measure exon levels and can thus be influenced by variables unrelated to RNA splicing, capture fundamentally different events." Suggest weaker wording (e.g. "mostly capture")

-"Islet sGenes were enriched in islet-specific co-expression networks (Figure 1j) and included numerous genes with well-established roles in islet biology and diabetes, including major drug targets (GLP1R, DPP4, ABCC8), regulators of hormone secretion..." Were these categories enriched? Show p-value

-Re fig 1a: Can you show imputation data/rna qc data? Did you account for ethnicity?

-Re fig 1a: Pancreas diagram in 1a very small suggest increasing size or amend/remove

-Color-to-dataset relationship unclear in 1d-i even if we examine 1a

-2b percentage should be 100 not 0 to 1.0

-2c unclear what color refers to what dataset

-2d do a p-value test, show more categories (e.g. FORGE2/GARFIELD analysis)

-2e unclear what "phenotypes" means. Also list top tissues

-"These observations, therefore, suggest that splicing variation in human islets could drive a component of T2D genetic susceptibility." You suggest an association with these traits but I suggest to rephrase to avoid "driving" as this is causal language

-"We reasoned that if splicing variation is truly instrumental in disease susceptibility at specific loci, a fraction of genetic signals for sQTLs and T2D association should show robust colocalization, and this could in turn point to specific transcripts underlying disease pathophysiology." Truly instrumental is strong wording.. implies causality. Also suggest rephrase "transcripts" to "candidate transcripts" (not that the transcripts themselves are candidates but rather they are candidates regarding underlying disease pathophysiology).

-"This identified 27 genes (42 splicing events) showing significant sTWAS with T2D risk, and 29 genes with eTWAS after multiple testing correction (Bonferroni $p= 8.6 \times 10^{-6}$ and 1.8×10^{-5} after correcting for 5,804 splicing junctions and 2,851 genes, respectively)" Is this TWAS a genome/transcriptome wide analysis or a subset of the whole transcriptome? Can it be called transcriptome-wide, subset of transcriptome-wide, or are there any caveats?

-3d notches overlap, indicating non sig sQTL. What are the nominal and corrected sQTL p values for these sites

-3e/d What are future steps for fine mapping/evaluating these sites

-4b are these annotations for pancreas or across tissues? Some evaluation of the tissue specificity of these sQTLs should be performed e.g. with FORGE2 or GARFIELD

-4b comparison with closed regions is not an appropriate background. A background adjusted for multiple genomic biases (e.g. cg content, tss distance, maf) should be applied (e.g. via FORGE2 or GARFIELD).

-4g locus and SNPs are not properly visible. Suggest enlarge or expand image

-"This convergence between GWAS and QTL fine-mapping provided further evidence that QTLs truly contain causal T2D risk variants." Truly, or "may truly"?

-Verb missing from second sentence of figure legend of figure 5.

-Last sentence of this figure legend needs to be rewritten as not clear.

-"This study adds splicing variation in pancreatic islets to the spectrum of molecular mechanisms that underlies T2D predisposition." underlies or "underly"?

- "T2D has a highly polygenic architecture, with a very large number of effector genes that individually exert small effects." effector or "of effector"?

- "we have nominated plausible candidate effectors from a tissue with strong disease-relevance" suggest remove hyphen

- "Raw RNA and genotypes will be made available from the European Genome-phenome Archive (EGA; <https://ega-archive.org/>). Full sQTL and eQTL results as well as variant effect predictions will be made available at <https://www.crg.eu/en/programmes-groups/ferrer-lab/datasets>" suggest add EGA accession number, and include summary stats and relevant data in Zenodo.

Response to reviewers

Atla et al, Genetics regulation of RNA splicing in pancreatic islets

We thank reviewers and the editor for a thorough assessment and many suggestions. We have addressed them as outlined below, and this has greatly improved the manuscript.

In addition to the responses to reviewers, we have now added analysis of sQTLs that co-localize with T1D association signals. There is growing evidence that beta cells are not innocent bystanders in the autoimmune destruction process. We highlight, for example, an interesting example of an sQTL in a cellular senescence gene. This observation supports recent work suggesting that intrinsic beta cell senescence processes promote autoimmune destruction in mouse and human models. We were initially hesitant to include this work because this signal is in LD with a major T1D risk association signal, but have since collaborated with John Todd and Dan Crouch who performed conditional analysis showing that our sQTL is an independent signal.

Another unsolicited novelty is that we realized that the CTRB2 “sQTL”, previously attributed by others to a caQTL, is actually an exon deletion. Understandably this came up as an exon skipping event in our analysis, but it is a fake sQTL. This same variant is associated with T1D, T2D, and PDAC, and shown to be a deletion in the context of PDAC studies. We mention this in the new version, but no longer highlight this as an sQTL proper.

Reviewer #1: *Understanding the role of splicing in pancreatic islets with respect to conferring risk for type 2 diabetes and related traits is a key endeavor. The authors appropriately analyzed islets sQTLs in data derived from 399 donors. Given that GTEx does not currently include splicing data for pancreatic islets, this data represents a particularly important resource for variant-to-function efforts. Here are my comments:*

1. In the Discussion, the authors state: "Splicing activity was estimated by junction usage, which does not directly inform isoform level regulation...". Given the authors are implicating a number of key events/genes, it would be optimal for a subset to be functionally validated/investigated with respect to isoform level - it would lend more credibility to the observations made.

We thank this reviewer for the supportive comments and constructive suggestions.

We agree that short read sequencing can provide information on alternative splicing variation, but it is more limited to predict full isoforms. We have acknowledged this in the **Discussion** (lines 411-413, page 10).

Given the inherent limitations of this data type, we have not attempted to validate predictions at full isoform level, since this would require using deep long read sequencing in a large number of samples, which would constitute a new project. RT-PCR is a theoretical possibility, but it would typically cover only parts of the transcript, since it is difficult to obtain quantitative information on alternate full length isoforms with this method.

In unpublished work, we have used PacBio RNA-seq on a small number of samples, at sufficient depth for annotating major isoforms, but not necessarily to accurately

quantify genetic effects. To reassure the reviewers that splice junction predictions are accurate, we show in **Appendix Figure 1 for reviewers** that PacBio full-length isoform sequencing data from human islets supports the predicted junctions from sQTL examples highlighted in the manuscript, namely all genes shown in **Figures 1d-i, 3d (SCAMP3)** and **4g (ERO1LB)**.

2. Another limitation is that pancreatic islets are a mixed cell population and thus the splicing observed really represents a composite of what is seen across the cell types present - it won't be until such work is carried out in the single cell context that one would be able to assign a splicing event to a given cell type. And of course, splicing is a dynamic process so one only gets a snapshot at a given static time point. The authors are encouraged to mention these points in the Discussion.

We agree and have thus expanded the **Discussion** (lines 417-420, page 10) to further emphasize the relevance of examining genetic effects on the appropriate context, whether is in the most appropriate cell type, upon response to external stimuli or during development.

3. The authors state that "Focusing on genes expressed in >10% of samples in each cohort..." was how they went about their analyses. This leads to a bias toward more abundant transcripts, and conversely could lead to the missing of potentially informative less abundant transcripts. The authors should describe why they selected this threshold.

We selected genes that have at least 5 mapped reads in 10% of the samples (40 out of 399 samples) for eQTL analysis. For sQTLs, we used junctions that have at least 5 junction-spanning reads in 10% of the samples. The rationale behind these filters was to boost power for mapping QTLs with high confidence. We agree that this has biased our results towards highly expressed genes across most samples, but we prioritized the identification of high confidence signals.

We are uploading unfiltered raw counts in our EGA submission, which can be used by others to study less abundant transcripts.

4. On page 7, the word 'robust' is used in rapid succession. How is this defined in this particular context?

Thank you, we have avoided overusing this term which is often dispensable.

Reviewer #2: *Atla and colleagues have carried out a large expression and splicing QTL study in human pancreatic islets. They have obtained new associations, they have integrated these data with GWAS data for relevant diseases and traits and they have uncovered interesting new biological associations. This paper is of broad interest to others in the field.*

While this study is properly conducted and advances new ground, I have a few comments, mostly methodological.

We thank this reviewer for careful and constructive comments and suggestions.

Main:

-Re fig 1 (and other parts of the ms): Enrichment was computed via enrichR/GREGOR did you try other methods to account for bias in background (e.g. FORGE2, GARFIELD)? I am concerned the p-values are a bit high and would like to see additional methods accounting for bias in enrichment.

We have redone the analysis for **Figure 2d** using GARFIELD, which yielded analogous results. This is now shown in **Supplementary Figure 2f**, **Supplementary Table 5** and described in **Methods**.

The results with both methods are summarized below

Figure 2 for reviewers (see also Supplementary Figure 2f). Enrichment of sQTL and eQTL variants in different functional genomic annotations using GREGOR and GARFIELD (using QTL p-value threshold of 5×10^{-5}). The x-axis represents GREGOR fold change of observed vs. expected number of SNPs or the GARFIELD odds ratio (OR) of the enrichment at each functional annotation. The dotted line represents 1.5-fold change, whiskers show the 95% CI of the GARFIELD OR. Two asterisks indicate significant GARFIELD enrichments after considering the number of effective annotations, the p-value thresholds used and the molecular QTL traits.

These results are persistent regardless of the QTL p-value threshold chosen (see **Appendix Figure 2 and Appendix Table 1 for reviewers**).

-Re fig 3a: Grey random background set area seems truncated on the right and does not cover blue spot x axis position.

We agree that this requires some clarification. The grey shaded area comprises 1000 p-value distributions based on newly generated control-sQTL variants, and thus, matches the number of observed sQTL variants. The number of eQTLs and sQTLs plotted in the QQ-plot are not the same. We now explain this in the legend and use a vertical dashed line to highlight that 1000 p-value control distributions match the sQTLs.

Figure 3 for reviewers (see also Figure 3a in the main manuscript). Quantile-quantile (QQ) plot showing observed T2D association p-values in human islet sQTLs (orange dots) and eQTLs (blue dots) against p-values under the null hypothesis. The grey shaded region represents 1000 p-value distributions (in the $-\log_{10}$ scale) of random sets of control sQTL variants (see Methods). Each set of control variants matches the number of islet sQTLs plotted.

"We thus compiled 106 putative effector genes for T2D or glycemic traits (FG, FI) reported here as well as previously reported co-localizing islet eQTLs10. This list excluded those exclusively detected in non-endocrine cells after analysis of single cell RNA-seq datasets (Supplementary Table 10), as well as non-coding transcripts and genes without known function." Why this extra non-endocrine filter?

The rationale behind this particular analysis is to focus on biological pathways, which requires studying various genes that are presumably expressed in the same cells.

However, we agree that exocrine or endothelial genes, for example, are potentially equally relevant. That is why we list all candidate effector genes identified in this study, along with the gene expression activity in pancreatic islet cell types (including genes only expressed in non-endocrine cells), in **Supplementary Table 12**.

Minor:

abstract: "This data exposes" (shouldn't this be "These data expose ")

We corrected this.

Intro:

"pointing to additional poorly understood genetic mechanisms." Is splicing poorly understood?

We have modified as follows:

*"A large fraction of T2D risk loci, however, cannot be ascribed to transcriptional regulatory mechanisms in pancreatic islets or other tissues, pointing to additional **non-***

coding molecular mechanisms that remain to be defined”.

Main text:

-"This overlap increased to 90% in comparisons with unpublished human islet transcript annotations built with long-reads (GA, AB, unpublished) (Supplementary Figure 2b). Can you add a preprint?"

The full manuscript is still in final stages of preparation, although the annotation used for this comparison is final. The comparison described here is a minor aspect of this manuscript so we hope that this will not represent a major obstacle. We hope to have the preprint in Biorxiv within the next two months.

-"This indicates that sQTLs, which directly measure splice junction variation, and exon-QTLs, which measure exon levels and can thus be influenced by variables unrelated to RNA splicing, capture fundamentally different events." Suggest weaker wording (e.g. "mostly capture")

We corrected the text exactly as suggested.

-"Islet sGenes were enriched in islet-specific co-expression networks (Figure 1j) and included numerous genes with well-established roles in islet biology and diabetes, including major drug targets (GLP1R, DPP4, ABCC8), regulators of hormone secretion..." Were these categories enriched? Show p-value

We provide enrichment p-values of sGenes among islet-specific co-expression networks in **Figure 1j**.

We noticed that many of the genes with well-established roles in beta cell biology or diabetes harbor sQTLs. The analysis of popular annotation programs (e.g., EnrichR) captured some of these genes in enriched annotations, but not necessarily all, and not necessarily in the most meaningful functional categories. We felt that the best way to highlight such genes, all of which are quite obviously important in diabetes, is to simply perform a manual curation. In fact, sQTLs do not specifically target genes that are known to be important, but it is still worth highlighting that many important genes have sQTLs.

We now state in the legend that this is a manually curated list "The top panel shows a manually curated list of examples with known functions in islet function and diabetes". We also provide in **Supplementary Table 3** citations of pivotal works that showcase the role of these genes in islet biology and diabetes.

-Re fig 1a: Can you show imputation data/rna qc data? Did you account for ethnicity?

Figure 4 for reviewers. Boxplot representation of the imputation information score calculated using QCTOOL for lead eQTLs and sQTLs identified at $FDR \leq 1\%$.

We calculated the imputation information score using QCTOOL for lead eQTLs and sQTLs identified in this study ($FDR \leq 1\%$). The imputation information score reflects the level of uncertainty in the imputed genotype data; a value of 0 indicates complete uncertainty in the predicted genotypes for a given SNP. In contrast, a value of 1 shows no uncertainty in the imputed genotype data, indicating that the genotypes for a given SNP are almost equivalent to observed genotype data. Our lead eQTL and sQTLs show median imputation information scores ~ 0.99 indicating that our results are not biased by imputation artifacts. This now is included in the manuscript as **Supplementary Figure 2a**.

Regarding ethnicity, as provided in **Supplementary Figure 1c-d**, we used samples largely clustered with individuals of European descent from 1000 Genomes.

Figure 5 for reviewers (based on Supplementary Figure 1c-d). Principal component (PC) analysis based on genotype data in 399 qualifying human islet samples. (c) PCs calculated from genotypes. Islet sample donors (light blue) were positioned according to PC1 (x-axis) and PC2 (y-axis) calculated from 1000 Genomes Phase 3 genotypes. (d) Differences in population structure according to genetic PC1 (x-axis) and PC2 (y-axis) between the four cohorts included in our panel of 399 islet transcriptomes.

As explained in **Methods**, we used our principal component analysis of genetic data, which captures population structure as well as other hidden batch effects, to correct QTL analysis (4 genetic PC components). Likewise, eQTL and sQTL analysis were

also corrected for 15 PCs and 10 PCs obtained from gene expression and splicing ratios, respectively. See methods and **Supplementary Figure 1 a-b**.

-Re fig 1a: Pancreas diagram in 1a very small suggest increasing size or amend/remove

We have deleted the islet diagram since it does not seem to be helpful.

-Color-to-dataset relationship unclear in 1d-i even if we examine 1a

We now provide a legend with color codes in **Figure 1a**.

-2b percentage should be 100 not 0 to 1.0

We thank the reviewer for pointing this out, we have changed to percentage

-2c unclear what color refers to what dataset

We have added a legend in **Figure 1a**.

-2d do a p-value test, show more categories (e.g. FORGE2/GARFIELD analysis)

Following the previous suggestion (see above and **Supplementary Figure 2f**), we now included GARFIELD in order to test enrichment of eQTLs and sQTLs among in genomic annotations taking into account genomic biases (MAF, LD and distance to the TSS).

-2e unclear what "phenotypes" means. Also list top tissues

We now clarified the label of the y-axis (e.g. *eGenes / Junctions (%)*), as well as the figure legend:

"Percentage of eGenes and Junctions with e and sQTLs at $FDR \leq 1\%$, respectively, shared in different number of GTEx tissues. We used significant eQTLs and sQTLs identified in 47 distinct GTEx V8 release tissues".

We also now include in **Supplementary Table 1** and **2** the GTEx tissues for which a given islet eGene or Junction also show a significant eQTL or sQTL signal, respectively.

-"These observations, therefore, suggest that splicing variation in human islets could drive a component of T2D genetic susceptibility." You suggest an association with these traits but I suggest to rephrase to avoid "driving" as this is causal language

We have omitted the term "drive", and state that this association "suggests that splicing could contribute to T2D" as follows:

"These observations, therefore, suggest that splicing variation in human islets could contribute to T2D genetic susceptibility".

-"We reasoned that if splicing variation is truly instrumental in disease susceptibility at specific loci, a fraction of genetic signals for sQTLs and T2D association should show robust colocalization, and this could in turn point to specific transcripts underlying disease pathophysiology." Truly instrumental is strong wording.. implies causality. Also

suggest rephrase "transcripts" to "candidate transcripts" (not that the transcripts themselves are candidates but rather they are candidates regarding underlying disease pathophysiology).

We are not stating that splicing is truly instrumental, but rather hypothesize that if that is the case, there should be colocalization (if there is no colocalization it is unlikely that they are causal, which is different from saying that colocalization proves causality).

We rephrased as follows (see parts in bold):

"We reasoned that if splicing variation **is linked to** disease susceptibility at specific loci, a fraction of genetic signals for sQTLs and T2D association should show **high colocalization evidence (posterior probability of shared association between both phenotypes ≥ 0.8)**, and this could in turn point to specific **candidate effector transcripts underlying disease pathophysiology**".

-*"This identified 27 genes (42 splicing events) showing significant sTWAS with T2D risk, and 29 genes with eTWAS after multiple testing correction (Bonferroni $p = 8.6 \times 10^{-6}$ and 1.8×10^{-5} after correcting for 5,804 splicing junctions and 2,851 genes, respectively)" Is this TWAS a genome/transcriptome wide analysis or a subset of the whole transcriptome? Can it be called transcriptome-wide, subset of transcriptome-wide, or are there any caveats?*

For this analysis, we follow to the terminology used in the original FUSION paper (Gusev, A. *et al. Nat Genet* (2016)). FUSION first identifies the fraction of the transcriptome that shows a significant *cis*-heritable component. This is a genome-wide analysis. Next, the expression or the splicing of genes with a *cis* genetic component is tested for association with a given GWAS trait. Thus, by focusing on the genetic component of the molecular phenotype (e.g., expression or splicing), FUSION avoids any association that is not mediated by genetic variation. Nonetheless, this does not preclude, according to the authors' nomenclature, the genome-wide basis of the association test between the *cis* heritable component of gene expression, or alternative splicing, and disease susceptibility or trait variation.

-*3d notches overlap, indicating non sig sQTL. What are the nominal and corrected sQTL p values for these sites*

We have now clarified in the figure legend whether the permutation p-values of the box-plots are significant at $FDR \leq 1\%$ and we point to **Supplementary Table 1** or **2** for eQTLs and sQTLs, respectively.

The summary statistics data for the *SCAMP3* splicing junction is provided in **Supplementary Table 2** for lead sQTL results (permutation pass QTLtools results at $FDR \leq 1\%$) and is as follows

1:155230241:155231877:clu_3220_NA_ENSG00000116521

Lead sQTL = **1:155209360:C:T**

Nominal P-value= 1.05059×10^{-9}

Beta = 0.00505287

Permutation Pass P-value = 5.16118×10^{-7} , qval = 8.661177×10^{-6}

Our lead sQTL for the TWAS analysis is **1:155203060:G:A**, that was identified as a nominally significant sQTL (summary statistics data in **Supplementary Data 2**):
Nominal P-value = 7.60287×10^{-9}
Beta = 0.00472951

We would like to remind the reviewer that Junction PSI or gene expression values used in any of the boxplots from this manuscript do not remove any additional confounding variation (e.g. adjustment for gene expression/splicing PCs or genetic PCs).

-3e/d What are future steps for fine mapping/evaluating these sites.

This study provides a resource to test novel mechanistic hypotheses for candidate GWAS effector genes. The next steps, in our mind, is to use appropriate model systems to explore candidate mechanisms, and to seek orthogonal types of evidence (e.g. rare coding mutations or otherwise). We have emphasized this in the discussion.

-4b are these annotations for pancreas or across tissues? Some evaluation of the tissue specificity of these sQTLs should be performed e.g. with FORGE2 or GARFIELD

We state in the legend that these are islet annotations.

Regarding the suggestion of the reviewer to evaluate the tissue specificity of sQTLs; we already examined in **Figure 2e** cross-tissue QTL activity of eGenes and Junctions showing eQTL and sQTL effects at FDR 1%.

We performed GREGOR (**Figure 2e**) and now GARFIELD (**Supplementary Figure 2f**) enrichments that show sQTLs enriched in 3' and 5' splice sites, for which tissue-specific annotations are not available (in contrast to epigenomic ones). Importantly, islet-specific RNA-binding protein (RBP) annotations are not widely available. This is a limitation to perform the analysis requested.

We nevertheless observed that enhancer and promoter annotations active in human pancreatic islets are the most enriched, most obviously in eQTLs (as expected) but also in sQTLs (see graph below). The enrichment for sQTLs could be indirect, due to LD with nearby splice variants and/or because of known role of transcription on splicing. Human pancreatic islet enhancer and promoter annotations were generated in Miguel-Escalada, I. et al (2019), whereas equivalent epigenomic annotations in 55 other tissues and cell-types were obtained from EpiMap (Boix CA. et al. 2021) (**Appendix Table 2**).

Given the limitations discussed above we have opted to leave this graph for reviewers rather than including in the manuscript. See also **Appendix Figures 3-5** and **Appendix Table 3** for reviewers.

Figure 6 for reviewers. Enrichment of sQTL (right) and eQTL (left) variants in promoter and enhancer annotations mapped in human pancreatic islets (Miguel Escalada et al) and other 55 EpiMap tissues. The x-axis represents the GARFIELD odds ratio (OR) of the enrichment at each tissue. Whiskers show the 95% CI of the GARFIELD OR.

-4b comparison with closed regions is not an appropriate background. A background adjusted for multiple genomic biases (e.g. cg content, tss distance, maf) should be applied (e.g. via FORGE2 or GARFIELD).

We would like to clarify that **Figure 4b** is not an enrichment analysis. This figure illustrates that fine-mapped eQTLs with largest posterior probability, without considering functional priors, tend to fall in open chromatin regions rather than in close chromatin. This suggests a concordance between genetic and functional prioritization of candidate causal eQTLs.

-4g locus and SNPs are not properly visible. Suggest enlarge or expand image

We have increased the image as suggested.

-"This convergence between GWAS and QTL fine-mapping provided further evidence that QTLs truly contain causal T2D risk variants." Truly, or "may truly"?

We agree with the reviewer' suggestion. We rephrased as follows:
"This convergence between GWAS and QTL fine-mapping provided further evidence that QTLs may contain causal T2D risk variants."

-Verb missing from second sentence of figure legend of figure 5. This has been corrected.

-Last sentence of this figure legend needs to be rewritten as not clear.

The text has been corrected as follows:

*“(b) Manual curation was used to illustrate **the relationship between components of the G protein-mediated insulinotropic pathway** targeted by islet QTLs linked to T2D and related traits .”*

-"This study adds splicing variation in pancreatic islets to the spectrum of molecular mechanisms that underlies T2D predisposition." underlies or "underly"?

We corrected this.

-"T2D has a highly polygenic architecture, with a very large number effector genes that individually exert small effects." effector or "of effector"?

We corrected this.

-"we have nominated plausible candidate effectors from a tissue with strong disease-relevance" suggest remove hyphen

We corrected this and rephrased it.

-"Raw RNA and genotypes will be made available from the European Genome-phenome Archive (EGA; <https://ega-archive.org/>). Full sQTL and eQTL results as well as variant effect predictions will be made available at <https://www.crg.eu/en/programmes-groups/ferrer-lab#datasets>" suggest add EGA accession number, and include summary stats and relevant data in Zenodo.

Following the reviewer suggestion, we uploaded summary statistics data in Zenodo (lead eQTL/sQTL variants, significant variant-to-gene/junction pairs, fine-mapped eQTLs and sQTLs). See the following link: <https://zenodo.org/record/6546807>. An EGA submission has been made. We will provide an EGA accession number shortly once the genotypes, the raw FASTQ files and raw unfiltered counts upload has been completed and validated by the EGA team.

Second round of review

Reviewer 1

The authors have satisfied the initial concerns of this reviewer.

Reviewer 2

The authors have addressed all of the reviewer comments.